# Hypothalamic representation of the imminence of predator threat detected by the vomeronasal organ in mice

Quynh Anh Thi Nguyen[1], Andrea Rocha[1,2], Ricky Chhor[2], Yuna Yamashita[2], Christian Stadler[1], Crystal Pontrello[2], Hongdian Yang[1,2], Sachiko Haga-Yamanaka[1,2]*

[1]Neuroscience Graduate Program, University of California, Riverside, Riverside, United States; [2]Department of Molecular, Cell and Systems Biology, University of California, Riverside, Riverside, United States

*For correspondence:
sachikoy@ucr.edu

Competing interest: The authors declare that no competing interests exist.

**Abstract** Animals have the innate ability to select optimal defensive behaviors with appropriate intensity within specific contexts. The vomeronasal organ (VNO) serves as a primary sensory channel for detecting predator cues by relaying signals to the medial hypothalamic nuclei, particularly the ventromedial hypothalamus (VMH), which directly controls defensive behavioral outputs. Here, we demonstrate that cat saliva contains predator cues that signal the imminence of predator threat and modulate the intensity of freezing behavior through the VNO in mice. Cat saliva activates VNO neurons expressing the V2R-A4 subfamily of sensory receptors, and the number of VNO neurons activated in response to saliva correlates with both the freshness of saliva and the intensity of freezing behavior. Moreover, the number of VMH neurons activated by fresh, but not old, saliva positively correlates with the intensity of freezing behavior. Detailed analyses of the spatial distribution of activated neurons, as well as their overlap within the same individual mice, revealed that fresh and old saliva predominantly activate distinct neuronal populations within the VMH. Collectively, this study suggests that there is an accessory olfactory circuit in mice that is specifically tuned to time-sensitive components of cat saliva, which optimizes their defensive behavior to maximize their chance of survival according to the imminence of threat.

## eLife Assessment

This **valuable** study addresses one way in which animals identify predator-associated cues and respond in a manner that reflects the imminence of the potential threat. The report shows that, in mice, fresh saliva from a natural predator (cat) elicits a greater defensive response compared to old cat saliva and implicates the vomeronasal organ and ventromedial hypothalamus as part of a circuit that underlies this process. The evidence supporting the main conclusions is **solid**. This study will be of interest to those interested in aversive behavior, its processes, and mechanisms.

## Introduction

Prey animals make defensive behavioral decisions through prompt and careful analysis of their environment when encountering predators. Immediacy of the threat, determined by perceived distance from predators, is one of the key factors influencing defensive behavioral decisions, which regulates the intensity of behavioral outputs (*Figure 1A*; *Blanchard and Blanchard, 1989*). Predator-derived sensory signals, or predator cues, are detected through multiple sensory systems, including the visual system, auditory system, and olfactory system. These individual sensory cues are sufficient to induce

innate defensive behavior such as fear-related freezing and flight, as well as anxiety-related risk assessment behavior. In nocturnal animals including rodents, olfaction is one of the major sensory modalities by which predator-derived chemical cues are detected (*Apfelbach et al., 2005*). Rodents respond to a variety of predator-derived chemical cues, such as odors from fox feces, snake skin, cat fur, and cat collars, in both the field and laboratory environments (*Apfelbach et al., 2005*; *Kats and Dill, 1998*; *Papes et al., 2010*; *Pérez-Gómez et al., 2015*; *Takahashi, 2014*). However, how animals use chemosensory cues to make appropriate behavioral decisions is not well understood.

The accessory olfactory system is one of the major sensory input channels through which predator-derived chemical cues are detected. It has been shown that some molecules derived from predator species activate sensory neurons in the vomeronasal organ (VNO) and induce defensive behavior (*Isogai et al., 2011*; *Papes et al., 2010*; *Pérez-Gómez et al., 2015*; *Samuelsen and Meredith, 2009*; *Tsunoda et al., 2018*). Moreover, a number of studies have revealed main components of the sensory neural circuits that mediate behavioral responses to chemical predator cues in rodents (*Blanchard et al., 2003*; *Canteras, 2002*; *Choi et al., 2005*; *Dielenberg and McGregor, 2001*; *Markham et al., 2004*; *Petrovich et al., 2001*). Predator signals detected by the VNO are transferred to the accessory olfactory bulb (AOB). The output neurons in the AOB (the mitral cells) then send the signals to the bed nucleus of the stria terminalis (BNST) and the posteroventral part of medial nucleus of the amygdala (MeApv), which connects with the dorsomedial part of the ventromedial hypothalamus (VMHdm) that contains neural populations directly controlling defensive behavioral outputs (*Gross and Canteras, 2012*; *LeDoux, 2012*; *Takahashi, 2014*). It is, therefore, reasonable to hypothesize that the VNO detects predator cues and sends sensory signals to the VMH, triggering appropriate defensive behavioral decisions in rodents. However, how the sensory signals detected through the VNO-to-VMH circuitry modulate behavioral decisions in specific contexts remains elusive.

In this study, we investigated sensory-to-hypothalamic neural circuitries that regulate defensive behavior induced by non-volatile chemical cues in cat saliva with different freshness. We examined the role of the VNO in behavioral responses and sought sensory receptor genes for the predator cues. Moreover, by comparing downstream neural populations activated by cat saliva with different freshness, we investigated how the imminence of predator threat is processed in the accessory olfactory circuitry to regulate behavioral outputs.

## Results

### Predator cues in cat saliva induce context-dependent defensive behavior in mice

To explore whether mice exhibit different intensities of defensive behavior in response to chemosensory predator cues with varying threat immediacy, we examined their response to cat saliva with different degrees of freshness, which potentially represents immediacy of the predator threat. Cat saliva has been considered as a source of predator cues found on cat fur and collars, which induce defensive behaviors in rodents (*Engelke et al., 2021*; *Papes et al., 2010*). Fresh saliva is assumed to indicate a more immediate threat to the prey animal, which likely triggers stronger defensive behavior. We exposed mice to a cotton swab containing cat saliva in their home cage where they know a flight route is not available (*Figure 1B*). We found that mice approached and sniffed the cotton swab, regardless of the presence of saliva, as part of their exploratory behavior toward a novel object in their home cage (*Videos 1–3*, *Figure 1C*; red boxes indicate the first contact). The latency to the first contact was slightly longer for saliva collected immediately before introduction to the mouse cage (fresh saliva) compared to that for saliva collected a few hours prior to the behavior test (old saliva) (*Figure 1D*). The difference in latencies was statistically significant, indicating that mice can perceive the change of cat saliva over time (*Figure 1D*). Baseline freezing episodes were observed in all groups of mice when the sample was introduced into the cage due to the experimenter's intervention (*Figure 1C*). However, the groups exposed to cat saliva continued to exhibit freezing behavior throughout the 10 min observation period, while mice exposed to a clean swab did not exhibit freezing behavior after the first contact (*Figure 1C and E*). Moreover, we found a correlation between the freshness of cat saliva and duration of freezing behavior in mice. Mice exposed to fresh saliva exhibited considerably longer freezing episodes, accounting for over 50% of the total duration of observation, whereas those exposed to old saliva exhibited freezing episodes for less than 20% of the total duration on average

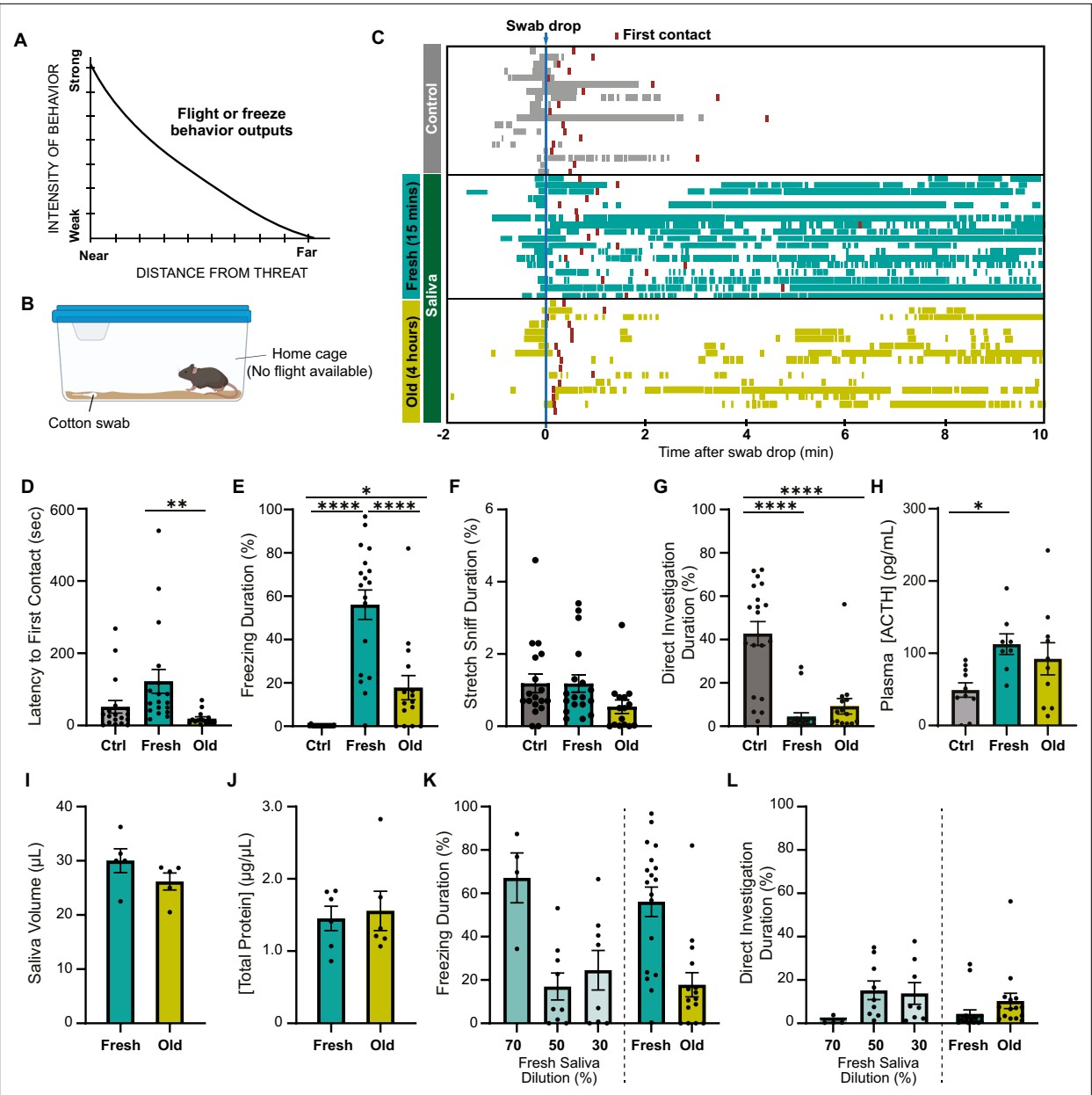

**Figure 1.** Predator cues in cat saliva induce context-dependent defensive behavior in mice. (**A**) A schematic diagram illustrating the elicitation of 'freeze' or 'flight' defensive behaviors in relation to distance. Adapted with permission from *Blanchard and Blanchard, 1989*. (**B**) A schematic illustration depicting a test chamber for evaluating defensive behaviors. (**C**) Raster plots displaying freezing episodes of individual mice exposed to a clean control swab (gray, n = 18), fresh saliva (green, n = 18), or old saliva (yellow, n = 15). The introduction of a clean, fresh, or old saliva swab into a mouse's cage is denoted as time 0. Red lines indicate the first contact with a swab for each mouse. (**D**) Latency to the first episode of contact with a swab. One-way ANOVA (F(2,48) = 5.166, p=0.0093) with Tukey's multiple-comparisons test (Ctrl vs. Fresh, p=0.0778; Ctrl vs. Old, p=0.5907; and Fresh vs. Old, p=0.0089). (**E**) The percentage of total freezing episodes after the first contact. One-way ANOVA (F(2,48) = 34.16, p<0.0001) with Tukey's multiple-comparisons test (Ctrl vs. Fresh, p<0.0001; Ctrl vs. Old, p=0.0468; and Fresh vs. Old, p<0.0001). (**F**) The percentage of total stretch sniff episodes. One-way ANOVA (F(2,48) = 2.307, p=0.1144). (**G**) The percentage of total direct investigation episodes. One-way ANOVA (F(2,48) = 25.17, p<0.0001) with Tukey's multiple-comparisons test (Ctrl vs. Fresh, p<0.0001; Ctrl vs. Old, p<0.0001; and Fresh vs. Old, p=0.5762). (**H**) Plasma ACTH concentrations following exposure to clean (n = 10), fresh (n = 8), or old saliva (n = 10) swabs. One-way ANOVA (F(2,25) = 3.735, p=0.0381) with Tukey's multiple-comparisons test (Ctrl vs. Fresh, p=0.0370; Ctrl vs. Old, p=0.1609; and Fresh vs. Old, p=0.6804). (**I**) Volume of saliva in individual swabs at 0 hr (Fresh) and 4 hr (Old) after collection. Two-tailed t-test (t = 1.415, df = 8, p=0.1949). (**J**) Concentrations of total proteins in fresh and old saliva. Two-tailed t-test (t = 0.3252, df = 10, p=0.7518). (**K**) The percentage of total freezing episodes induced by diluted fresh cat saliva (70%, n = 4; 50%, n = 9; 30%, n = 8), compared to the ones induced by fresh and old saliva. One-way ANOVA (F(4, 47) = 0.6604, p<0.0001) with Tukey's multiple-comparisons test. Statistical results are reported in *Supplementary file 1*. (**L**) The percentage of total direct investigation episodes to diluted fresh cat saliva (70%, n = 4; 50%, n = 9; 30%, n = 8), compared

*Figure 1 continued on next page*

*Figure 1 continued*

to the ones induced by fresh and old saliva. One-way ANOVA ($F_{(4, 49)}$ = 2.272, p=0.0748). In (**D**--**L**), the values are presented as means ± S.E.M., with individual dots representing individual mice. *, **, and **** denote significance levels of p<0.05, 0.01, and 0.0001, respectively. Data for fresh and old saliva in (**K**) and (**L**) are reused from (**E**) and (**G**), respectively.

The online version of this article includes the following source data and figure supplement(s) for figure 1:

**Source data 1.** Data used to generate the graphs in *Figure 1*.

**Figure supplement 1.** Predator cues in cat saliva induce context-dependent defensive behavior in mice.

**Figure supplement 2.** Representative western blot analysis of Fel d 4 protein bands in fresh and old saliva samples.

**Figure supplement 2—source data 1.** Original file for western blot analysis displayed in *Figure 1—figure supplement 2*.

**Figure supplement 2—source data 2.** Original file for western blot analysis displayed in *Figure 1—figure supplement 2*, indicating the relevant bands and sample conditions.

**Figure supplement 3.** Recombinant Fel d 4 does not induce freezing behavior in mice.

**Figure supplement 3—source data 1.** Original file for SDS-PAGE gel displayed in *Figure 1—figure supplement 3*.

**Figure supplement 3—source data 2.** Original file for SDS-PAGE gel displayed in *Figure 1—figure supplement 3*, indicating the relevant bands and sample conditions.

(*Figure 1E*). We found no significant difference in the latency to the first freezing episode from the first contact with a swab between mice exposed to fresh and old saliva (*Figure 1—figure supplement 1A*). Mice also exhibited stretch sniffing risk assessment behavior, which did not differ toward any of the swab types (*Figure 1F*, *Figure 1—figure supplement 1B*). On the contrary, mice exposed to a swab containing either fresh or old saliva significantly avoided interacting with the swab, while mice exposed to a clean control swab spent a significant amount of time directly interacting with the swab to investigate it, including direct sniffing and chewing (*Figure 1G*). The comparison of temporal behavioral patterns revealed a slightly higher frequency of direct investigation behavior toward old saliva compared to fresh saliva at the beginning of the exposure period (*Figure 1—figure supplement 1C*). Additionally, the stress hormone adrenocorticotropic hormone (ACTH) in plasma showed a trend toward twofold upregulation after exposure to both fresh and old saliva (*Figure 1H*). These results suggest that mice can still recognize predator threat through old saliva, but freezing behavior in response to fresh saliva attenuates quickly over time.

Saliva-containing swabs were kept in sealed containers immediately after collection from cats to prevent evaporation, ensuring that the saliva volumes at the time of behavioral testing were indistinguishable between fresh and old samples (*Figure 1I*). The concentrations of total proteins in both fresh and old saliva were also indiscernible, which was approximately 1.5 µg/µL (*Figure 1J*). Furthermore, the amount of Fel d 4, one of the most abundant proteins in saliva, was nearly equivalent between fresh and old saliva (*Figure 1—figure supplement 2*), indicating that the differences between fresh and old cat saliva lie in specific components rather than the total or major saliva content. One possible explanation for this difference is the time-dependent reduction of freezing-inducing components in old saliva. Indeed, exposure to diluted fresh saliva resulted

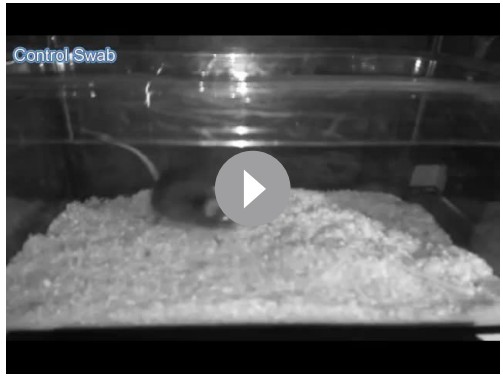

**Video 1.** Mouse behavior toward a control swab.
https://elifesciences.org/articles/92982/figures#video1

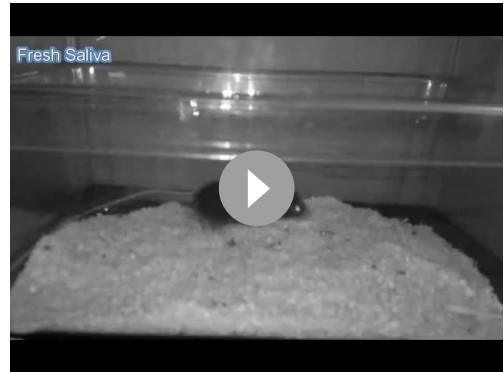

**Video 2.** Mouse behavior toward a swab containing fresh saliva.
https://elifesciences.org/articles/92982/figures#video2

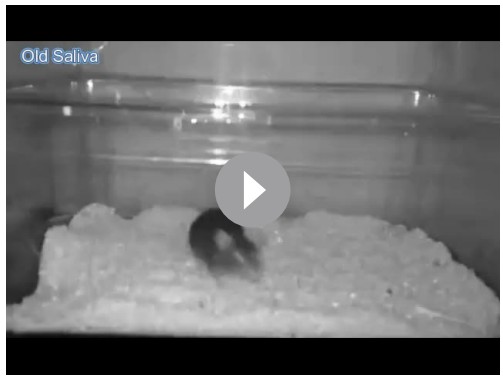

**Video 3.** Mouse behavior toward a swab containing old saliva.
https://elifesciences.org/articles/92982/figures#video3

in a shorter duration of freezing behavior. Fresh saliva diluted to 70% induced freezing behavior for a duration equivalent to that of undiluted fresh saliva, while freezing behavior in response to 50 and 30% fresh saliva was significantly reduced to the same duration as that observed with old saliva (*Figure 1K*). The duration of direct interaction with swabs containing 70% and 50–30% fresh saliva also exhibited a similar trend to that observed with fresh and old saliva swabs, respectively (*Figure 1L*).

Taken together, these results indicate that cat saliva contains predator cues that attenuate over time, which is detected by mice to assess the imminence of predator threat and modulate defensive behavioral outputs. More specifically, fresh saliva is recognized as an immediate danger and induces more robust freezing behavior, while old saliva is recognized as a less immediate danger and induces less robust freezing behavioral responses.

## Cat saliva activates VNO neurons and modulates freezing behavior

As the freezing behavior was induced when mice made direct contact with the saliva-containing swab, we postulated that direct contact with the predator cues in cat saliva is necessary for freezing behavior induction. To examine the necessity of direct contact, we placed a cotton swab with or without cat saliva into a porous container and investigated mouse's behavior toward the container. The pores on the container allow volatile chemicals in cat saliva to diffuse out, while nonvolatile chemicals stay inside of the container with the swab. Interestingly, we found that when we introduced a container containing a swab with fresh saliva into the home cage, the mice no longer exhibited freezing behavior toward the container, but instead directly investigated it similarly to the control swab-containing container (*Figure 2A and B*). This suggests that freezing behavior requires direct contact with the saliva-containing swab and that the freezing-inducing cues in cat saliva detected in this scheme are nonvolatile.

Mice have the ability to detect nonvolatile chemical cues through the VNO (*Mohrhardt et al., 2018*). Ethologically relevant chemical cues from different animals, such as pheromones and predator cues, are sucked up into the VNO and activate VNO sensory neurons when animals contact the chemical source (*Luo et al., 2003*). We, therefore, explored the role of the VNO in inducing freezing in response to cat saliva using a VNO-deficient laboratory mouse line that lacks the primary signal transduction ion channel in VNO neurons (transient receptor potential cation channel, subfamily C, member 2-knockout line or Trpc2-KO line) (*Leypold et al., 2002*). In Trpc2-KO mice, chemosensory signals detected by sensory receptor proteins are expected to be attenuated before being relayed to the brain due to the lack of the downstream transduction channel in VNO neurons (*Leypold et al., 2002*; *Liman et al., 1999*; *Stowers et al., 2002*). Thus, Trpc2-KO mice have been widely utilized to examine roles of the VNO in various behaviors (*Beny-Shefer et al., 2017*; *Ferrero et al., 2013*; *Haga et al., 2010*; *Kimchi et al., 2007*; *Leypold et al., 2002*; *Papes et al., 2010*; *Pérez-Gómez et al., 2015*; *Stowers et al., 2002*). Upon exposure to a fresh saliva-containing swab, Trpc2-KO mice failed to exhibit freezing behavior, in contrast to their wild-type littermates who displayed freezing responses. Additionally, the Trpc2-KO mice directly investigated the saliva swab with a similar level of intensity as the control swab, while the wild-type littermates actively avoided it (*Figure 2C and D*). These findings suggest that VNO activation, disrupted in Trpc2-KO mice, is necessary for the freezing behavior observed in response to cat saliva in mice.

To investigate if VNO neurons are indeed activated by cat saliva, we analyzed immunoreactivity (IR) of phosphorylated ribosomal protein S6 subunit, pS6, in the VNO of mice exposed to cat saliva. As the cellular pS6 level is greatly upregulated upon activation of VNO neurons, it has been used as a sensitive marker to examine neural activation in the VNO (*Carvalho et al., 2020*; *Itakura et al., 2022*;

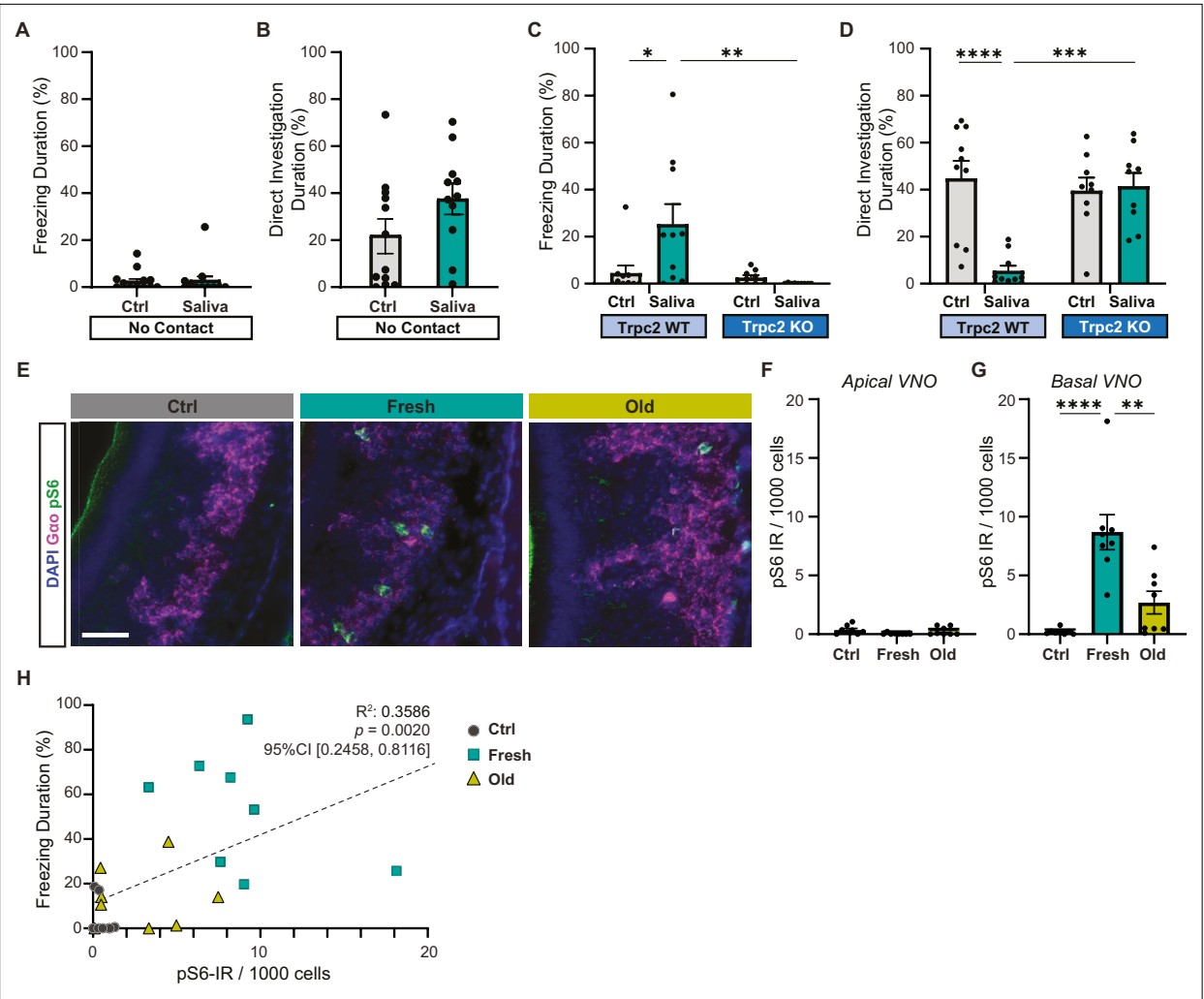

**Figure 2.** Cat saliva activates vomeronasal organ (VNO) neurons and modulates freezing behavior. (**A**) The percentage of total freezing episodes directed toward control (n = 12) or fresh saliva (n = 11) swabs in cassettes that prevent direct contact with the swabs. Two-tailed *t*-test ($t$ = 0.08745, df = 21, p=0.9311). (**B**) The percentage of total investigation episodes toward cassettes containing control or fresh saliva swabs. Two-tailed *t*-test ($t$ = 1.662, df = 21, p=0.1114). (**C**) The percentage of total freezing episodes directed toward control or fresh saliva swabs in Trpc2-WT (n = 10) or -KO (n = 9) mice. Two-way ANOVA ([genotype] $F_{(1, 34)}$ = 3.662, p=0.0641, [swab type] $F_{(1, 34)}$ = 8.053, p=0.0076, [genotype] and [swab type] $F_{(1, 34)}$ = 5.963, p=0.0200) with Tukey's multiple-comparisons test (WT-Ctrl vs. KO-Ctrl, p=0.9922; WT-Ctrl vs. WT-Saliva, p=0.0164; WT-Ctrl vs. KO-Saliva, p=0.9136; KO-Ctrl vs. WT-Saliva, p=0.0100; KO-Ctrl vs. KO-Saliva, p=0.9832; WT-Saliva vs. KO-Saliva, p=0.0037). (**D**) The percentage of total direct investigation episodes toward control or fresh saliva swabs in Trpc2-WT or -KO mice. Two-way ANOVA ([genotype] $F_{(1, 34)}$ = 11.46, p=0.0018, [swab type] $F_{(1, 34)}$ = 7.725, p=0.0088, [genotype] and [swab type] $F_{(1, 34)}$ = 13.96, p=0.0007) with Tukey's multiple-comparisons test (WT-Ctrl vs. KO-Ctrl, p=0.9051; WT-Ctrl vs. WT-Saliva, p<0.0001; WT-Ctrl vs. KO-Saliva, p=0.9732; KO-Ctrl vs. WT-Saliva, p=0.0006; KO-Ctrl vs. KO-Saliva, p=0.9949; WT-Saliva vs. KO-Saliva, p=0.0003). (**E**) Representative images illustrating the expression of pS6 (green) and Gα₀ (magenta) in the VNO of mice exposed to control, fresh, or old saliva swabs. DAPI was used as a nuclear counterstain. Scale bar: 50 μm. (**F, G**) The number of pS6-IR positive cells in 1000 VNO cells in the apical (**F**) and basal (**G**) regions of the VNO neuroepithelium in mice stimulated with control (n = 8), fresh saliva (n = 8), and old saliva (n = 8) swabs. (**F**) One-way ANOVA ($F_{(2, 21)}$ = 2.562, p=0.1010). (**G**) One-way ANOVA ($F_{(2, 21)}$ = 17.90, p<0.0001) with Tukey's multiple-comparisons test (Ctrl vs. Fresh, p<0.0001; Ctrl vs. Old, p=0.2262; and Fresh vs. Old, p=0.0014). (**H**) The percentage of total freezing episodes in relation to the number of pS6-positive cells in the VNO of individual mice (n = 8 per swab type). Correlation analysis was conducted using Spearman's rank correlation coefficient as total data points did not pass Shapiro–Wilk and Kolmogorov–Smirnov normality tests. Statistical results are reported in the graph. (**A–D, F, G**) the values are presented as means ± S.E.M., with individual dots representing individual mice. *, **, ***, and **** denote significance levels of p<0.05, 0.01, 0.001, and 0.0001, respectively.

The online version of this article includes the following source data for figure 2:

**Source data 1.** Data used to generate the graphs in *Figure 2*.

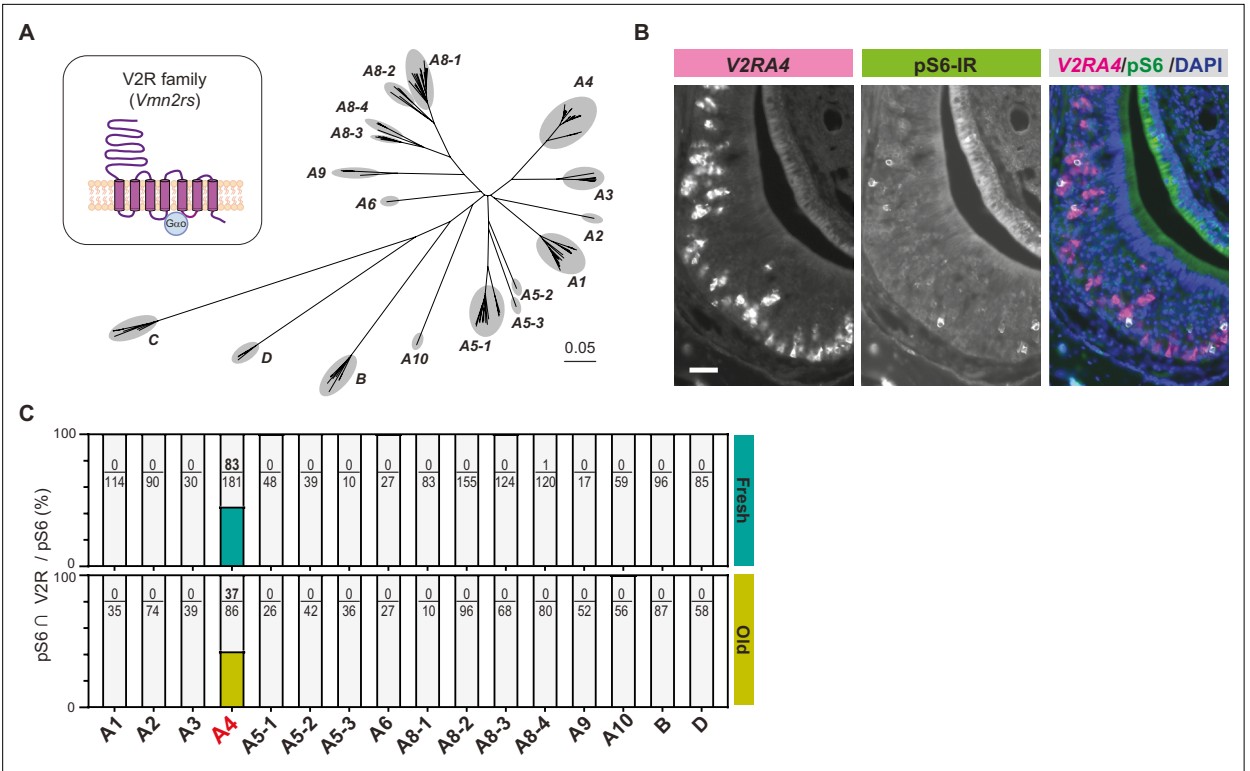

**Figure 3.** Cat saliva activates the V2R-A4 subfamily of the vomeronasal receptors. (**A**) Phylogenetic tree of the V2R family and its subfamilies used for V2R cRNA probe design. (**B**) Representative images displaying the expression of *V2RA4* (magenta) and pS6 (green) in the vomeronasal organ (VNO) of a mouse exposed to a fresh saliva swab. DAPI was utilized as a nuclear counterstain. Scale bar: 50 μm. (**C**) The percentage of cells co-labeled with V2R probes and anti-pS6 antibody among the total pS6-positive cells in the VNO of mice exposed to fresh saliva (top, from 3 to 6 animals) or old saliva (bottom, from 3 to 6 animals).

The online version of this article includes the following figure supplement(s) for figure 3:

**Figure supplement 1.** Representative images of *V2R* in situ hybridization and pS6 immunohistochemistry for each V2R subfamily.

*Osakada et al., 2022*; *Tsunoda et al., 2018*). We observed that exposure to both fresh and old cat saliva induced an increase in pS6-IR in the basal VNO neurons that express Gαo protein (*Figure 2E and G*). Conversely, pS6-IR was rarely observed in the apical VNO neurons (*Figure 2F*). Notably, we found that the number of pS6-positive neurons was more than three times higher in mice exposed to fresh saliva compared to that exposed to old saliva (*Figure 2G*). Strikingly, when the duration of freezing was plotted as a function of pS6-IR signals in the VNO of individual animals, we observed a significantly positive correlation ($R^2$ = 0.3586, 95% CI [0.2458, 0.8116], p=0.0020) between the numbers of active sensory neurons and behavioral outputs (*Figure 2H*), suggesting that more intense freezing behavior is induced when more VNO neurons are activated by cat saliva.

Taken together, these results indicate that freezing-inducing predator cues are detected by Gαo-positive sensory neurons in the VNO, and these sensory neurons convey the imminence of predator threat to higher brain centers.

## Cat saliva activates the V2R-A4 subfamily of the vomeronasal receptors

The significant reduction of the freezing duration to aged saliva reflects possible attenuation of signal inputs to a specific population of sensory receptors, or alternatively, it may reflect a change of sensory receptor repertoires due to qualitative changes of predator cues over time. To test these possibilities, we examined sensory receptors expressed in the neurons activated by fresh and old saliva. The basal VNO neurons express sensory receptors that belong to vomeronasal receptor type 2 (V2R) family of G protein-coupled receptors in mice (*Figure 3A*; *Mohrhardt et al., 2018*). The mouse V2R family consists of over 120 intact genes, which are divided into V2R A1-6, A8-10, and B-D subfamilies based on sequence homology (*Figure 3A*; *Francia et al., 2014*; *Silvotti et al., 2011*; *Silvotti et al., 2007*).

With the exception of the V2R C group that is co-expressed with other groups of V2Rs in the basal VNO neurons, each sensory neuron expresses only one member of the V2R receptor, allowing individual neurons to respond to specific molecules (*Mohrhardt et al., 2018*).

To investigate which subfamilies of V2R are expressed in cat saliva-activated VNO neurons, we designed in situ hybridization (ISH) probes that detect individual V2R subfamilies and performed dual staining of V2R ISH with pS6-IR (*Figure 3—figure supplement 1*). We found that the majority of pS6-IR signals induced by cat saliva co-localized with signals of the ISH probe against the V2R-A4 subfamily (*Figure 3B*, *Figure 3—figure supplement 1*). Approximately 45% of pS6-IR signals induced by fresh saliva (83 out of 181) co-localized with V2R-A4 probe signals, with less than 1% of pS6-IR (1 out of 120) co-localized with A8-4 probe signals (*Figure 3C*). Notably, V2R-A4 probe signals also co-localized with approximately 45% of pS6-IR signals induced by old saliva (37 out of 86). None of the other probe signals co-localized with pS6-IR induced by both fresh and old saliva. These results indicate that members of the V2R-A4 subfamily are potential sensory receptors for the predator cues contained in both fresh and old cat saliva, and varying levels of imminence of predator threat stimulate the same type of sensory receptors in a quantitatively different manner to regulate the freezing behavior.

## Cat saliva activates neurons in the AOB and defensive behavioral circuit

We next aimed to understand how the differential activation of sensory receptor neurons in the VNO by different freshness of cat saliva influences behavioral decisions through downstream neural circuits (*Figure 4A*). We investigated this by examining the induction of cFos, an immediate early gene product, in the projection neurons (mitral cells) and local inhibitory neurons (granule cells) in the AOB of mice exposed to cat saliva. The axons of apical and basal VNO neurons project to the anterior and posterior regions of the AOB (aAOB and pAOB), respectively (*Mohrhardt et al., 2018*). We found that the activation of the basal VNO neurons by cat saliva leads to predominant induction of cFos-IR signals in the pAOB, whose glomerulus layer is Gαi-IR negative (*Figure 4B–F*). The numbers of cFos-positive mitral and granule cells in the pAOB in saliva-exposed mice are significantly higher than those in mice exposed to the control swab (*Figure 4D and F*). However, contrary to our expectations, we did not observe any differences in the numbers of cFos-positive mitral and granule cells in the AOB between fresh and old saliva-exposed mice (*Figure 4D and F*). These results suggest that, although the AOB processes saliva signals from the VNO following predator cue detection, the relationship between the degree of AOB activation and freezing behavior intensity is nonlinear.

To further investigate how signals of fresh and old saliva are processed in the downstream circuitry, we examined induction of cFos in the neural substrates involved in innate defensive behaviors (*Figure 4A*). We investigated the BNST and MeApv, as these nuclei receive direct ascending inputs from the mitral cells in the AOB (*Mohrhardt et al., 2018*). We also investigated the VMH and dPAG, as these nuclei have been shown to control defensive behavioral outputs (*Silva et al., 2013*). Upon exposure to fresh and old saliva, cFos expression in the MeApv, VMH, and dPAG, but not in the BNST, appeared to be upregulated compared to the control stimulus (*Figure 4G–J*, *Figure 4—figure supplement 1*). Notably, cFos expression was significantly increased in the VMH of mice exposed to fresh and old saliva compared to the control (*Figure 4I*), consistent with previous observations that the dorsomedial and central subdivisions of VMH (VMHdm/c) contain subpopulations of neurons that control freezing behavior (*Engelke et al., 2021*; *Kunwar et al., 2015*; *Silva et al., 2016*; *Silva et al., 2013*; *Wang et al., 2015*). However, there was no difference in the number of cFos-positive cells between the two saliva-exposed groups in the neural substrates tested (*Figure 4G–J*). These results suggest that, although the defensive circuit is activated by cat saliva signals, the relationship between the degree of activation and freezing behavior intensity is nonlinear, similar to the situation in the AOB.

Taken together, these results indicate that the relationship between neural activity and behavioral output is more complex in the downstream circuit than in the VNO. More specifically, the numbers of active neurons and intensity of freezing behavior do not appear to correlate in the downstream regions. This suggests that fresh and old cat saliva may differentially activate functionally distinct neural populations within these regions, which in turn leads to quantitatively different freezing behavioral responses.

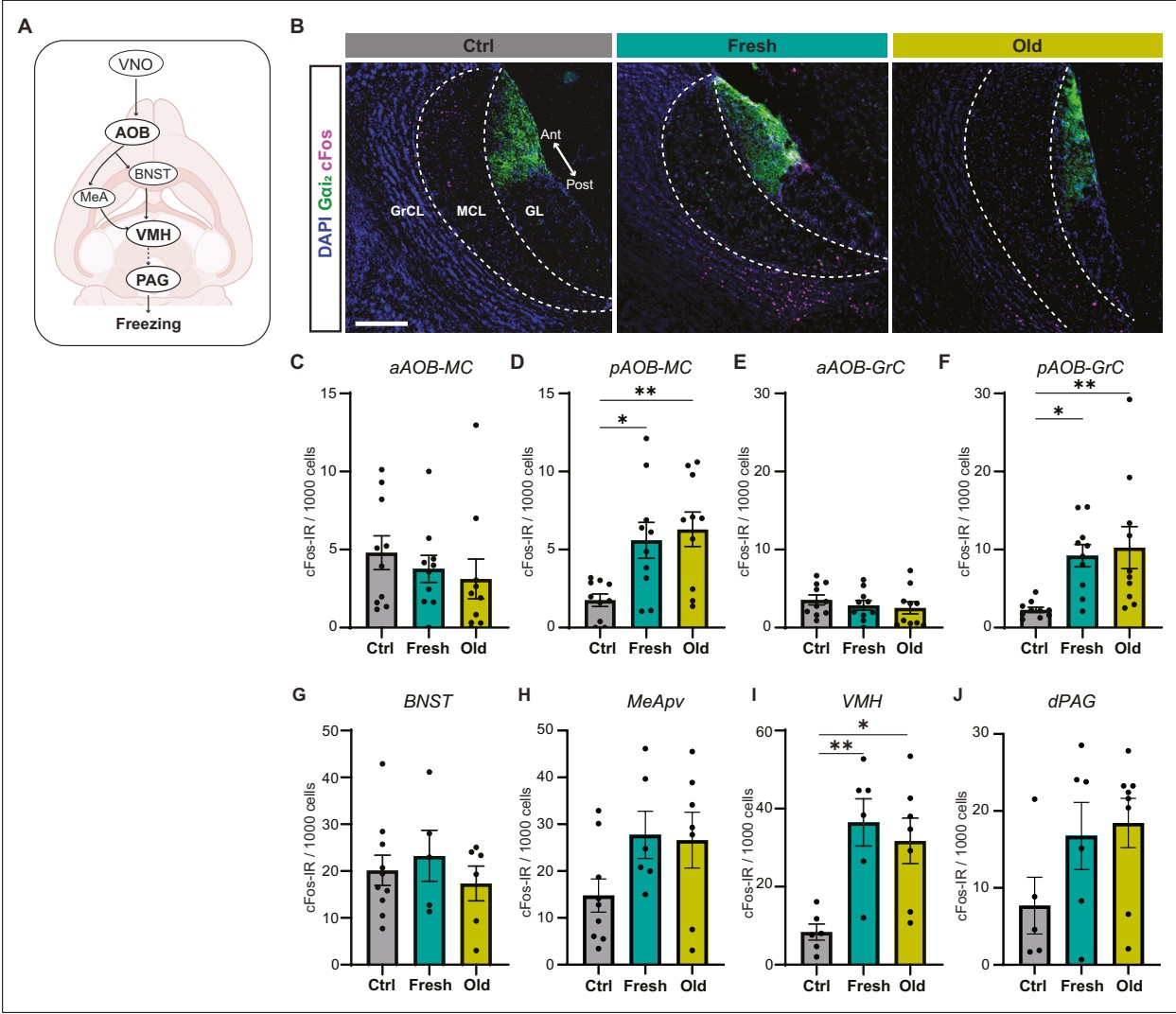

**Figure 4.** Cat saliva activates neurons in the accessory olfactory bulb (AOB) and defensive behavioral circuit. (**A**) Schematic diagrams illustrating the vomeronasal organ (VNO) sensory pathway that controls defensive behaviors. (**B**) Representative images displaying the expression of cFos (magenta) in the mitral cell (MCL) and granule cell (GCL) layers of the AOB of mice exposed to control, fresh, or old saliva swabs. G$\alpha_{i2}$ (green) visualizes the anterior region of the glomerular layer (GL). DAPI was utilized as a nuclear counterstain. Scale bar: 200 µm. (**C, D**) The number of cFos-IR positive mitral cells per 1000 mitral cells in the anterior (**C**) and posterior (**D**) regions of the AOB (n = 10 per swab type). (**C**) One-way ANOVA ($F_{(2, 27)}$ = 0.6106, p=0.5504). (**D**) One-way ANOVA ($F_{(2, 27)}$ = 6.665, p=0.0044) with Tukey's multiple-comparisons test (Ctrl vs. Fresh, p=0.0210; Ctrl vs. Old, p=0.0059; and Fresh vs. Old, p=0.8593). (**E, F**) The number of cFos-IR positive granule cells per 1000 granule cells in the anterior (**E**) and posterior (**F**) regions of the AOB (n = 10 per swab type). (**E**) One-way ANOVA ($F_{(2, 27)}$ = 0.5831, p=0.5651). (**F**) One-way ANOVA ($F_{(2, 27)}$ = 5.996, p=0.0070) with Tukey's multiple-comparisons test (Ctrl vs. Fresh, p=0.0262; Ctrl vs. Old, p=0.0099; and Fresh vs. Old, p=0.9105). (**G**) The number of cFos-IR positive neurons per 1000 cells in the BNST in mice stimulated with control (n = 10), fresh saliva (n = 5), and old saliva (n = 6) swabs. One-way ANOVA ($F_{(2, 18)}$ = 0.4375, p=0.6524). (**H**) The number of cFos-IR positive neurons per 1000 cells in the MeApv in mice stimulated with control (n = 9), fresh saliva (n = 6), and old saliva (n = 7) swabs. One-way ANOVA ($F_{(2, 19)}$ = 2.454, p=0.1128). (**I**) The number of cFos-IR positive neurons per 1000 cells in the VMH in mice stimulated with control (n = 6), fresh saliva (n = 6), and old saliva (n = 7) swabs. One-way ANOVA ($F_{(2, 16)}$ = 8.332, p=0.0033) with Tukey's multiple-comparisons test (Ctrl vs. Fresh, p=0.0043; Ctrl vs. Old, p=0.0127; and Fresh vs. Old, p=0.7880). (**J**) The number of cFos-IR positive neurons per 1000 cells in the dPAG in mice stimulated with control (n = 5), fresh saliva (n = 6), and old saliva (n = 8) swabs. One-way ANOVA ($F_{(2, 16)}$ = 2.136, p=0.1506). (**J**) The number of cFos-IR positive neurons per 1000 cells in the dPAG in mice stimulated with control (n = 5), fresh saliva (n = 6), and old saliva (n = 8) swabs. One-way ANOVA ($F_{(2, 16)}$ = 2.136, p=0.1506). In (**C–J**), the values are presented as means ± S.E.M., with individual dots representing individual mice. * and ** denote significance levels of p<0.05 and 0.01, respectively.

The online version of this article includes the following source data and figure supplement(s) for figure 4:

**Source data 1.** Data used to generate the graphs in *Figure 4*.

**Figure supplement 1.** Representative images displaying the expression of cFos (red) in the BNST, MeApv, VMH, and dPAG of mice exposed to control, fresh, or old saliva swabs.

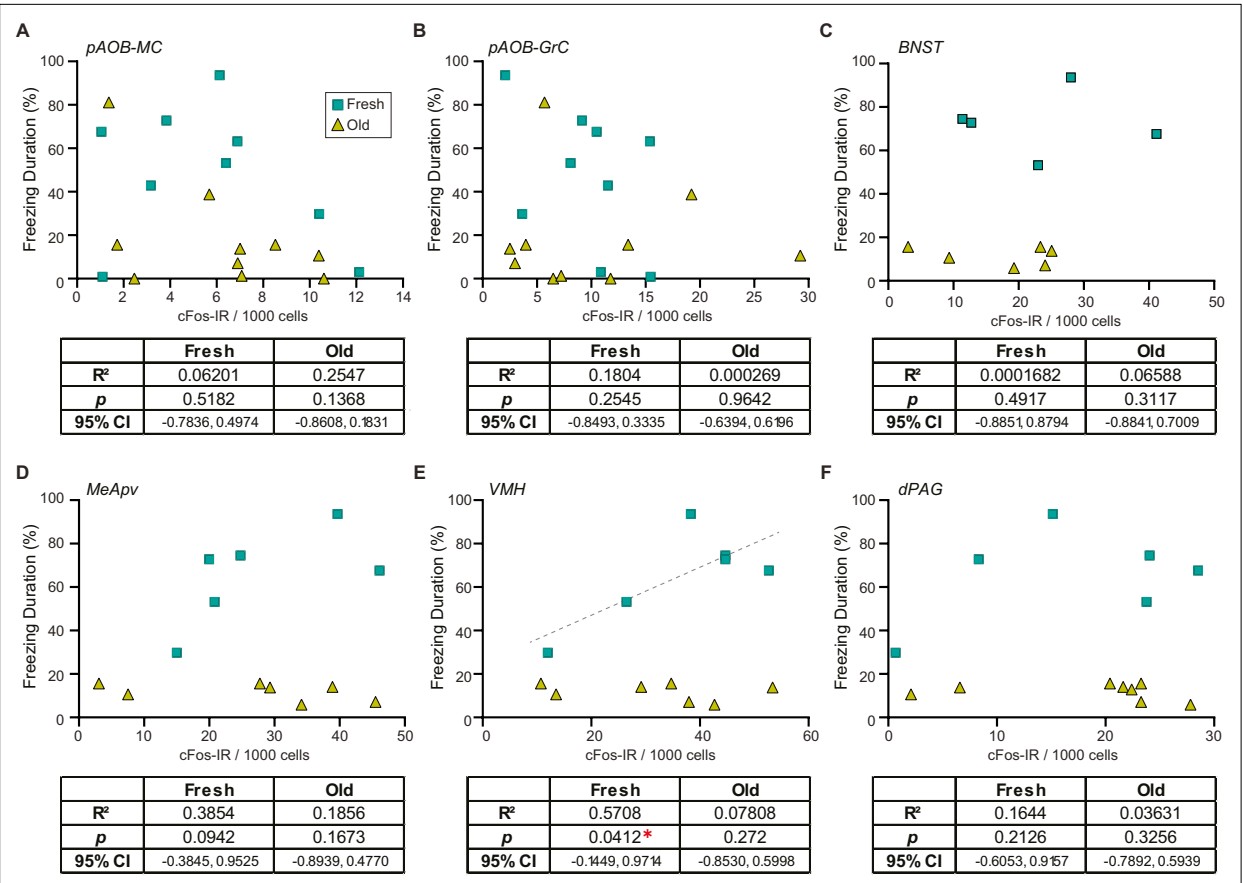

**Figure 5.** Fresh and old cat saliva differentially activate ventromedial hypothalamus (VMH) neurons to control defensive behavior. (A–F) Percentage of the total freezing episodes as a function of the number of cFos-positive posterior accessory olfactory bulb (AOB) mitral cells (A) and granule cells (B), BNST (C), MeApv (D), VMH (E), and dPAG (F) neurons. Green, and yellow symbols represent values from mice exposed to fresh saliva and old saliva, respectively. Sample size is same as seen in *Figure 4*. Correlation was analyzed using Pearson correlation coefficient as data points in each group pass Shapiro–Wilk and Kolmogorov–Smirnov normality tests. Statistical results are reported in the table below each graph. * denotes statistical significance with p<0.05.

The online version of this article includes the following source data for figure 5:

**Source data 1.** Data used to generate the graphs in *Figure 5*.

## Fresh and old cat saliva differentially activate VMH neurons to control defensive behavior

To investigate the potential differences between fresh and old saliva-activated neurons, we analyzed correlations of the duration of freezing behavior and numbers of cFos-IR signals in different cell types in individual animals from different exposure groups. To our surprise, we found that the numbers of cFos-IR signals in AOB mitral and granule cells, as well as those in neurons in the BNST and dPAG, did not correlate with the duration of freezing behavior in any of the exposure groups (*Figure 5A–C and F*). However, we observed a significant positive correlation between the number of cFos-IR signals and behavior intensity only in the fresh saliva-exposed group in the VMH ($R^2$ = 0.5708, 95% CI [–0.1449, 0.9714], p=0.0412) while no such correlation was observed in the old saliva-exposed group (*Figure 5E*). We observed a similar positive correlation trend in the MeApv ($R^2$ = 0.3854, 95% CI [–0.3845, 0.9525], p=0.0942), although it was not statistically significant possibly due to low sample numbers (*Figure 5D*). Taken together, these findings support our hypothesis that fresh and old saliva may differently activate the downstream circuit.

We therefore examined whether fresh and old cat saliva activate neurons in spatially distinct regions within the VMH. We observed the distribution of cFos-IR neurons in five serial brain sections from rostral to caudal VMH regions. The VMH region within each hemisphere section was further

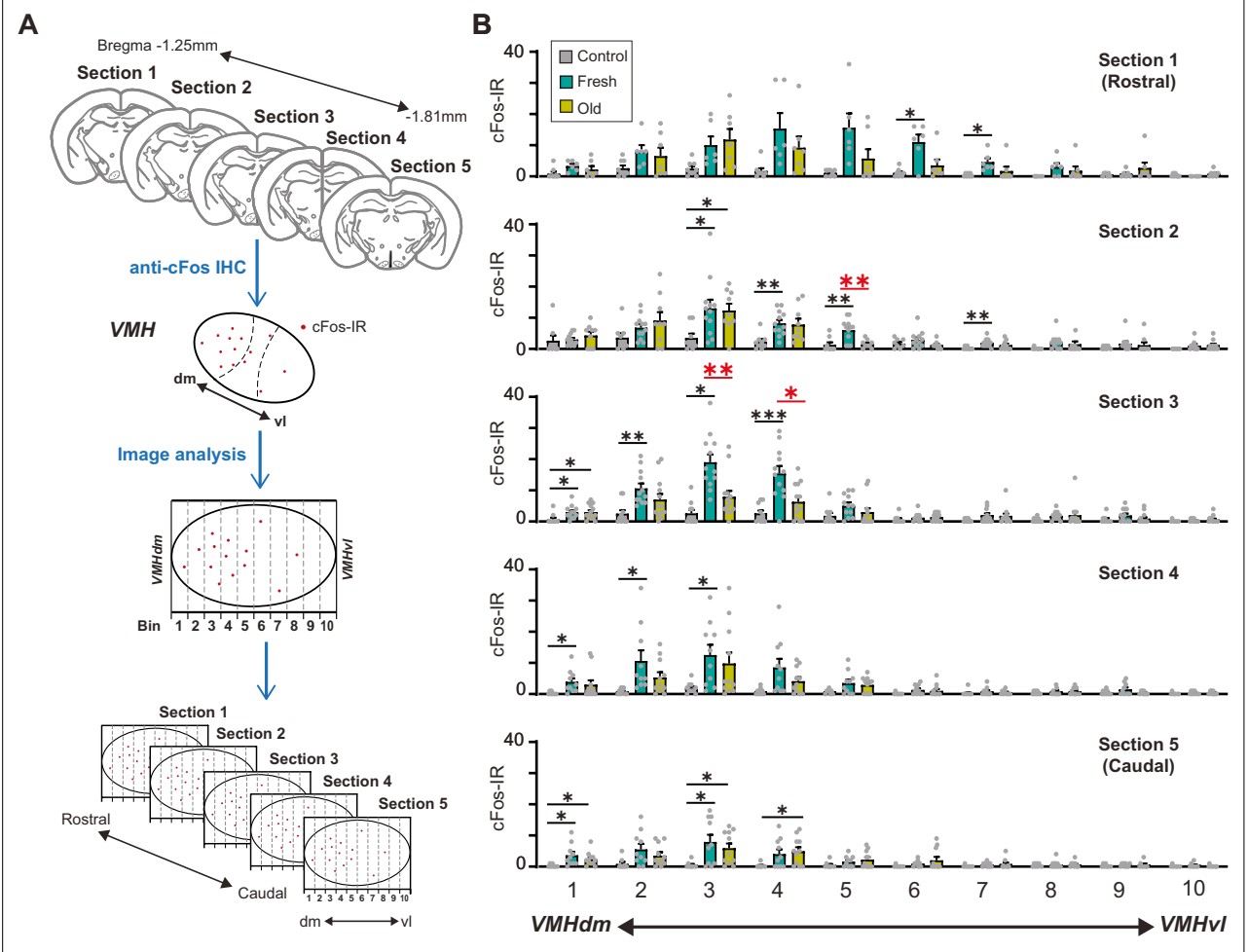

**Figure 6.** A population of ventromedial hypothalamus (VMH) neurons is more sensitive to fresh saliva than old saliva. (**A**) Schematic diagram illustrating the cFos-IR distribution analysis. Each VMH region in five consecutive sections (140 μm apart) was prepared along the dorsomedial (dm)-ventrolateral (vl) axis and divided into 10 bins that are vertical to the dm-vl axis. cFos-IR signals in each bin were counted. (**B**) The number of cFos-IR-positive cells in each bin in the VMH hemisphere sections. Sample images are the same as seen in **Figure 4** (n = 6–7 animals). The values are presented as means + S.E.M., with individual dots representing values from individual VMH hemisphere sections. Statistical analysis was performed using two-way ANOVA with Tukey's multiple-comparisons test. Statistical results are reported in **Supplementary file 1**. Black * and ** denote significance levels of $p<0.05$ and 0.01, respectively, between control and fresh or old saliva conditions. Red * and ** denote significance levels of $p<0.05$ and 0.01, respectively, between fresh and old saliva conditions.

The online version of this article includes the following source data for figure 6:

**Source data 1.** Data used to generate the graphs in **Figure 6**.

divided into 10 bins along the dorsomedial-ventrolateral axis, and cFos-IR signals in each bin were counted (**Figure 6A**). The majority of cFos-IR signals induced by both fresh and old cat saliva were localized in the dorsomedial to central part of the VMH (bins 1–5). Strikingly, the numbers of cFos-IR signals induced by fresh and old saliva were significantly different in the section 2 bin 5, as well as the section 3 bin 3 and bin 4, which are adjacent to each other (**Figure 6B**). The cFos-IR signals in the other regions did not show significant difference between fresh and old saliva-exposed mice. These results suggest that there is a distinct population of neurons clustered in a close proximity within the VMH, which is more sensitive to fresh saliva than old saliva. This population of neurons in the VMHdm may play a role in inducing robust freezing behavior to fresh cat saliva. Moreover, these results also suggest that the neurons that induce robust freezing to fresh saliva and less robust freezing to old saliva reside in the same region of the VMH and may use different mechanisms to encode freezing intensity.

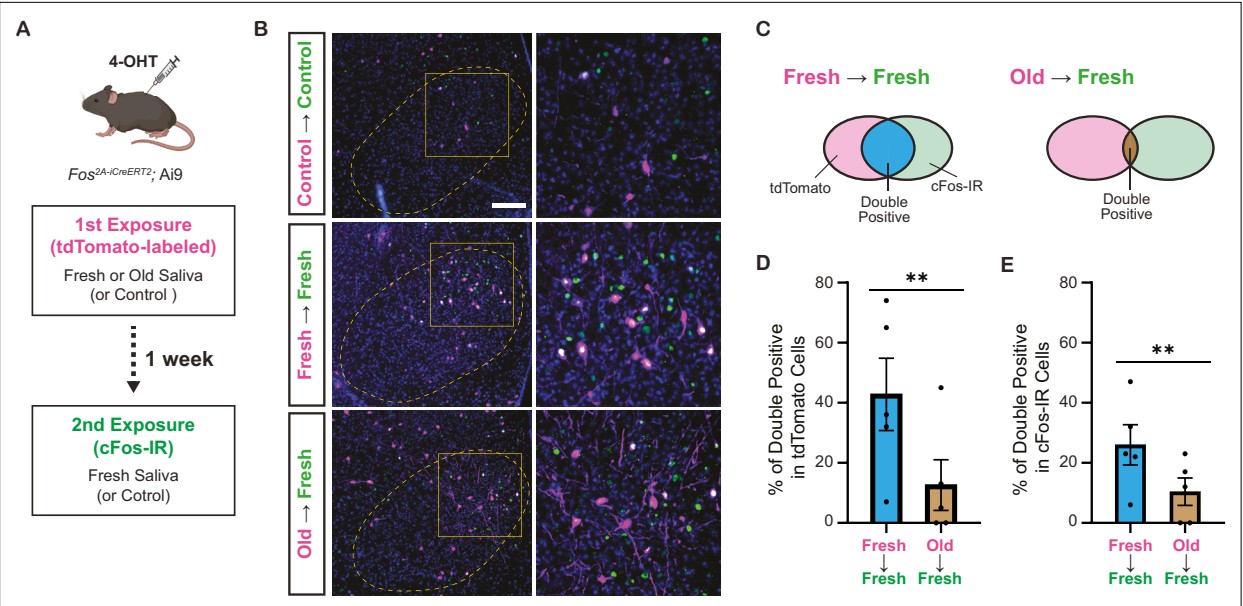

**Figure 7.** Fresh and old cat saliva activate largely different populations of ventromedial hypothalamus (VMH) neurons. (**A**) Schematic diagram illustrating the double exposure experiment using *Fos*[2A-iCreERT2]; Ai9 mice to visualize the populations of neurons activated by fresh and old saliva in the same individual mouse. (**B**) Representative images showing tdTomato expression (magenta) and cFos-IR (green) in the VMH of mice exposed to different swabs: Control (first and second), fresh saliva (first and second), or old saliva (first), and fresh saliva (second). Overlapping signals are indicated in white. DAPI was used for nuclear counterstaining. Yellow dotted lines outline the VMH. Areas enclosed by yellow squares in the left images are enlarged in the right images. Scale bar: 100 µm. (**C**) Schematic diagram illustrating the color code for the graphs in (**D**) and (**E**). (**D**) The percentage of double-positive cells among tdTomato-expressing cells in mice exposed to fresh (first) and fresh (second) saliva (n = 5) and old (first) and fresh (second) saliva (n = 5). A permutation test (p=0.0035). (**E**) The percentage of double-positive cells among cFos-IR-positive cells in mice exposed to fresh (first) and fresh (second) saliva (n = 5) and old (first) and fresh (second) saliva (n = 5). A permutation test (p=0.0060). In (**D**) and (**E**), the values are presented as means ± S.E.M., with individual dots representing individual mice. ** denotes a significance level of p<0.01.

The online version of this article includes the following source data for figure 7:

**Source data 1.** Data used to generate the graphs in *Figure 7*.

## Fresh and old cat saliva activate largely different populations of VMH neurons

Do fresh and old saliva activate distinct VMH neurons within the same individuals? To investigate this, we performed a double exposure experiment using *Fos*[2A-iCreERT2]; Ai9 mice (*Figure 7A*). The *Fos*[2A-iCreERT2] allele enabled the expression of iCre-ERT2 recombinase in activated neurons, facilitating *loxP*-mediated recombination in a 4-hydroxytamoxifen (4-OHT)-dependent manner (*Allen et al., 2017*; *DeNardo et al., 2019*). This recombination resulted in the expression of tdTomato from the Ai9 allele, allowing visualization of neurons activated during the first saliva exposure. One week after the initial exposure, the same mice were subjected to a second saliva exposure for 1 hr, leading to cFos expression induced by the neural activity triggered during the second exposure. The cFos protein was detected by immunohistochemistry (*Figure 7A*).

In animals exposed to both control (first) and control (second) swabs, sparse tdTomato and cFos-IR signals were observed in the VMH. However, in animals exposed to two saliva samples, substantial populations of neurons in the VMH expressed tdTomato and/or cFos-IR signals (*Figure 7B*). Quantification of tdTomato and cFos-IR double-positive cells among tdTomato-labeled cells (*Figure 7C*) revealed that 43% (mean per animal: 61/143) of the cells activated by fresh saliva during the first exposure were also activated by fresh saliva during the second exposure, whereas only 16% (17/106) of cells activated by old saliva during the first exposure were activated by fresh saliva during the second exposure (p=7.5e⁻⁶, chi-square test). The difference in the fraction of overlapping cells between fresh and old saliva exposures was maintained when we compared the two groups of animals (*Figure 7D*, p=0.0035, permutation test). Additionally, quantification of tdTomato and cFos-IR double-positive cells among cFos-IR cells indicated that 27% (61/226) of cells activated by fresh saliva during the

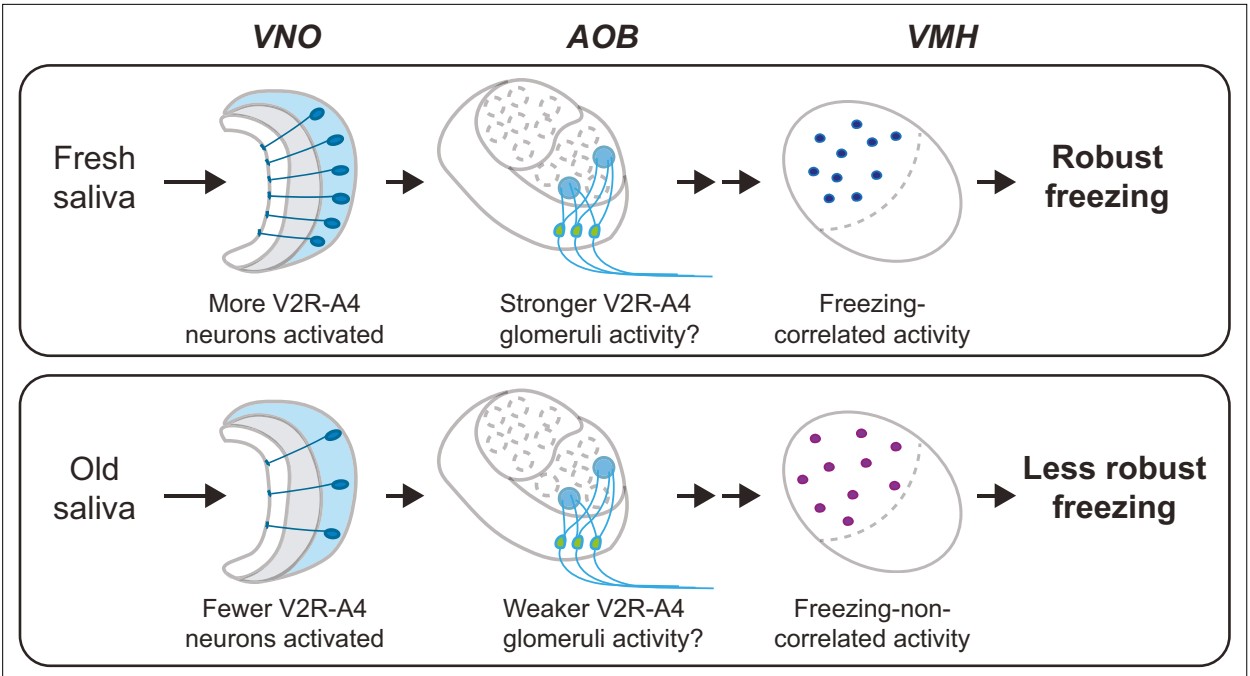

**Figure 8.** A model of the vomeronasal organ to ventromedial hypothalamus (VNO-to-VMH) pathway processing imminence of predator threat. The circuitry is selectively activated based on the varying imminence of predator cues. Fresh saliva activates more V2R-A4-expressing VNO neurons (blue), which potentially results in stronger activation of the V2R-A4 glomeruli. This eventually triggers freezing-correlated neural activity in the VMH. In contrast, old saliva activates fewer V2R-A4-expressing VNO neurons (blue), which potentially results in weaker activation of the V2R-A4 glomeruli, leading to freezing-non-correlated activity in the VMH.

second exposure were previously activated by fresh saliva, whereas 15% (17/112) of cells activated by fresh saliva during the second exposure were previously activated by old saliva (p=0.015, chi-square test). The difference in the fraction of overlapping cells between fresh and old saliva exposures was also significant in this analysis (*Figure 7E*, p=0.0060, permutation test). Together, these results demonstrate that fresh and old cat saliva activate largely different populations of neurons within the VMH.

Overall, our results suggest that the qualitative properties of predator signals, such as freshness, can induce distinct neural processing and ultimately lead to different defensive behavioral outcomes.

## Discussion

This study provides evidence for the role of cat saliva as a source of predator cues that signal imminent predator threat and regulate the intensity of freezing behavior through the accessory olfactory pathway in mice (*Figure 8*). Notably, both fresh and old saliva activate neurons expressing the V2R-A4 subfamily of sensory receptors, suggesting the existence of a specific receptor group responding to the predator cues. Importantly, the number of vomeronasal neurons activated in response to saliva correlates with the freshness of saliva and the intensity of freezing behavior. Moreover, the number of VMH neurons activated by immediate threat (fresh saliva) exhibits a positive correlation with the intensity of behavioral outputs, suggesting their direct involvement in the modulation of freezing behavior. In contrast, the activation of VMH neurons by less immediate threat (old saliva) does not show such a correlation, suggesting the presence of distinct populations of neurons in the VMH processing imminence of predator threat. Consistent with this, we detected a specific population of VMH neurons that are more sensitive to fresh saliva than to old saliva. We also showed that fresh and old saliva activate largely different populations of VMH neurons in the same individuals, further suggesting the existence of distinct populations of VMH neurons that modulate defensive behavior according to the imminence of threat.

## The VNO as the sensor of predator cues that induce fear-related behavior

This study demonstrates that the VNO is involved in detecting and processing chemosensory predator signals that induce fear-related freezing behavior. Previous studies have shown that the VNO mediates defensive behaviors against predators that are rather related to anxiety state, such as risk assessment, area avoidance, and reduced exploratory and locomotion behaviors (*Isogai et al., 2011*; *Papes et al., 2010*; *Pérez-Gómez et al., 2015*; *Tsunoda et al., 2018*). The difference between these studies and our current study is likely due to different sample types and/or their freshness, as we observed that fear-related behavior is correlated with freshness of predator samples. It is interesting to assume that the biological sources of predator cues, such as saliva, urine, and tears, contain varieties of unidentified cues that convey time-sensitive signals through the mouse accessory olfactory system. This is consistent with the fact that VNO neurons expressing more than half of vomeronasal receptors and a significant fraction of AOB mitral cells have been shown to be activated by predator-derived samples (*Ben-Shaul et al., 2010*; *Isogai et al., 2011*). Therefore, it is important to investigate how the sensory pathway in the accessory olfactory system processes varieties of predator cues, which eventually leads to defensive behavioral decisions in higher brain centers.

Moreover, our study indicates that time-dependent alterations of the chemosensory predator signals can be detected by the VNO, which induces altered defensive behavioral responses. For instance, 4-hr-old saliva induces reduced freezing behavior compared to fresh saliva. On the other hand, duration of direct investigation remains low toward both fresh and old saliva, and the stress hormone ACTH level is upregulated in both cases. A future study ought to identify specific molecules—most likely proteins or peptides—in cat saliva responsible for inducing freezing behavior in mice. A cat lipocalin allergen, Fel d 4, has been identified as a trigger for risk assessment defensive behavior in mice (*Papes et al., 2010*). While Fel d 4 appears to be a potential candidate for inducing freezing behavior, our analysis showed that the amount of Fel d 4 did not differ between fresh and old saliva (*Figure 1—figure supplement 2*). Additionally, we found that recombinant Fel d 4 protein did not induce freezing behavior (*Figure 1—figure supplement 3*). Therefore, it is unlikely that Fel d 4 alone is the factor responsible for triggering freezing behavior. This suggests the possibility of an unidentified molecule or a combination of multiple unidentified molecules contributing to this response. Once these molecules are identified, it will be crucial to investigate how their quantity and quality change over time and how these changes correlate with freezing behavior. Such an investigation will offer valuable insights into how mice perceive the immediacy of a threat through these chemical signals.

## V2R-A4 subfamily as the receptor for predator cues in cat saliva

Screening of sensory receptors in the VNO neurons activated by fresh and old saliva demonstrated that members of the V2R-A4 subfamily are the candidate receptors for the predator cues in cat saliva. The V2R-A4 subfamily has been shown to be expressed in the VNO neurons that are activated by biological samples derived from various mammalian predators (*Carvalho et al., 2020*; *Isogai et al., 2011*; *Rocha et al., 2024*; *Tsunoda et al., 2018*). Therefore, it is likely that specific members of the V2R-A4 subfamily are the receptors for the predator cues in cat saliva that induce freezing behavior.

Dual staining of vomeronasal receptors with immediate early gene products has been effectively utilized to identify receptors for several pheromones (*Carvalho et al., 2020*; *Haga et al., 2010*; *Isogai et al., 2018*; *Isogai et al., 2011*; *Itakura et al., 2022*; *Kimoto et al., 2005*; *Osakada et al., 2022*; *Osakada et al., 2018*; *Tsunoda et al., 2018*). However, due to very high sequence homologies among members of the V2R-A4 subfamily (*Francia et al., 2014*; *Rocha et al., 2024*), alternative approaches need to be considered to further isolate the receptors responsible for the detection of predator cues in cat saliva. Potential approaches include receptor mRNA profiling with pS6 immunoprecipitation from the VNO of mice stimulated by cat saliva (*Isogai et al., 2018*; *Jiang et al., 2015*), as well as receptor mRNA profiling from isolated single cells activated by cat saliva in GCaMP imaging using the VNO slices in vitro (*Haga-Yamanaka et al., 2014*; *Wong et al., 2020*). Receptor candidates identified using either of the methods can be further confirmed by examining necessity and sufficiency for detecting cat saliva using genetically modified mouse lines (*Haga et al., 2010*; *Haga-Yamanaka et al., 2014*; *Isogai et al., 2018*; *Itakura et al., 2022*; *Osakada et al., 2022*; *Osakada et al., 2018*; *Tsunoda et al., 2018*).

## Differential processing of fresh and old saliva signals in the VNO-to-VMH pathway

Our correlation analyses between the number of activated neurons and behavior intensity suggest that sensory signals encoding different imminence of predator threat are differentially processed in the VNO-to-VMH pathway. The differential behavioral responses do not appear to result from global neuromodulations of the neural circuit by stress hormones as differential endocrinological responses were not observed (*Figure 1H*). This suggests that more specific neural processing happens in the circuitry. In the circuitry downstream of the VNO, sensory signals can converge and diverge from the AOB before reaching the VMH. The axon terminals of hundreds of VNO neurons expressing the same receptor converge into a few specific glomerulus structures (*Belluscio et al., 1999*; *Del Punta et al., 2002*; *Rodriguez et al., 1999*), and, in some cases, neurons expressing receptors in the same subfamily send axons into the same glomeruli (*Wagner et al., 2006*). Individual AOB mitral cells extend multiple primary dendrites into multiple glomeruli (*Imamura et al., 2020*; *Larriva-Sahd, 2008*; *Takami and Graziadei, 1991*; *Wagner et al., 2006*; *Yonekura and Yokoi, 2008*). In turn, efferent axons of the mitral cells project to multiple brain substrates, namely the BNST, the MEA, and the posteromedial cortical amygdaloid nucleus (PMCo) (*Davis et al., 1978*; *Von Campenhausen and Mori, 2000*). These structural features indicate that sensory inputs to the AOB mitral cells are converged, and outputs from the mitral cells may be diverged.

These features of the ascending neural pathway within the AOB may contribute to the sorting of sensory signals derived from fresh and old saliva. Fresh saliva activates more V2R-A4-expressing VNO neurons than old saliva, which may result in stronger activation of V2R-A4 glomeruli in fresh saliva-exposed mice compared to those exposed to old saliva. We assume that such differential activations of V2R-A4 glomeruli between fresh and old saliva result in the differential activation of VMH neurons (*Figure 8*). The methodology that we undertook in this study, cFos-immunohistochemistry, only provides information about neurons responding to cat saliva above a given threshold. It is therefore important to examine how the neurons in this receptor circuitry respond to fresh and old saliva using more quantitative techniques, such as in vivo electrophysiological recording, in future studies.

## Differential activation of VMH neurons potentially underlying distinct intensities of freezing behavior

Our findings indicate that the imminent predator cues present in cat saliva activate a specific subpopulation of VMH neurons, whose overall activity positively correlates with the intensity of freezing behavioral outputs (*Figure 5E*). In contrast, the less imminent predator cues activate a subpopulation of VMH neurons, whose overall activity does not correlate with the intensity of freezing behavior (*Figure 5E*). These findings suggest that induction of robust freezing behavior is determined by a distinct set of VMH neurons, and the intensity of freezing behavior is encoded separately, possibly by the intensity or dynamics of the neuronal firing, which cannot be detected by the cFos-IR used in this study. The VMHdm/c region contains several subpopulations of neurons that coordinate various defensive behavioral responses (*Engelke et al., 2021*; *Esteban Masferrer et al., 2020*; *Kennedy et al., 2020*; *Silva et al., 2016*; *Tobias et al., 2023*). Previous studies have demonstrated that optogenetic stimulation of VMHdm/c neurons induces freezing/immobility responses, suggesting that these neurons mediate freezing behavior in response to predator cues (*Kunwar et al., 2015*; *Wang et al., 2015*). In the current study, the distribution of cFos-IR in the VMH revealed a population of neurons clustered in close proximity that are more sensitive to fresh saliva than old saliva (*Figure 6B*). Therefore, we propose that imminent predator cues in cat saliva may activate these specific neurons with certain excitation levels to directly induce robust freezing behavior, while less imminent predator cues in cat saliva may use a different mechanism to regulate freezing behavior.

What distinguishes the subpopulation of VMHdm/c neurons activated by fresh saliva from those activated by old saliva? One possibility is that these neurons stimulate distinct downstream neural circuitries. The VMHdm/c neurons project to two major nuclei, the anterior hypothalamic nucleus (AHN) and the periaqueductal gray (PAG), which directly influence different aspects of defensive behavior (*Canteras et al., 1994*; *Wang et al., 2015*). Specifically, selective activation of the VMHdm/c-to-PAG pathway has been shown to cause freezing/immobility behavior, while activation of the VMHdm/c-to-AHN pathway rather promotes avoidance and induces immobility to a lesser extent (*Wang et al., 2015*). Although the number of cFos-positive cells in the dPAG did not seem to correlate with the

intensity of freezing behavior in any saliva exposure group in our study (*Figure 5F*), it is possible that fresh saliva-activated neurons send inputs to the PAG, whereas old saliva-activated neurons send inputs to the AHN and/or other projection sites. Thus, the number of old saliva-activated cells in the VMH may correlate with different aspects of defensive responses, rather than freezing behavior.

In summary, our study proposes a predator defensive circuitry that involves the accessory olfactory system and the VMHdm/c. Understanding the precise neural mechanisms of how different degrees of imminence of predator threat are processed in this circuit will provide valuable insights into the decision-making processes underlying defensive behaviors in animals.

# Materials and methods

## Key resources table

| Reagent type (species) or resource | Designation | Source or reference | Identifiers | Additional information |
|---|---|---|---|---|
| Genetic reagent (*Mus musculus*) | *Trpc2* knock-out | **Leypold et al., 2002** | | Backcrossed for >7 generations with C57BL/6N |
| Genetic reagent (*M. musculus*) | *Fos^{2A-iCreERT2}* | The Jackson Laboratory | #030323 | Backcrossed for >7 generations with C57BL/6N |
| Genetic reagent (*M. musculus*) | Ai9 | The Jackson Laboratory | #007909 | |
| Strain, strain background (*M. musculus*) | C57BL/6N | Charles River | # 027 | |
| Strain, strain background (*Felis catus*) | Domestic shorthair | Nutrition and Pet Care Center at the University of California, Davis | | |
| Antibody | Anti-Fel d 4 (rabbit polyclonal) | Inbio | PA-FD4 | WB (1:1000) |
| Antibody | Anti-pS6 (rabbit polyclonal) | Cell Signaling Technology | #4858 | IHC (1:200) |
| Antibody | Anti-Gαi2 (mouse monoclonal) | Cell Signaling Technology | #4858 | IHC (1:500) |
| Antibody | Anti-cFos (guinea pig polyclonal) | Synaptic Systems | #226 308 | IHC (1:1000) |
| Antibody | Horseradish peroxidase-conjugated anti-rabbit IgG | Cell Signaling Technology | #7074 | WB (1:5000) |
| Antibody | Anti-rabbit IgG, Alexa Fluor 647 | Life Technologies | #a21245 | IHC (1:500) |
| Antibody | Anti-mouse IgG2b, Alexa Fluor 546 | Life Technologies | #a21143 | IHC (1:500) |
| Antibody | Anti-guinea pig IgG, Alexa Fluor 647 | Life Technologies | #a21450 | IHC (1:500) |
| Antibody | Anti-DIG-POD | MilliporeSigma | #11207733910 | ISH (1:200) |
| Sequence-based reagent | pMAL-c2x-Fel d 4 | **Papes et al., 2010** | | |
| Sequence-based reagent | pET-28a(+)-Fel d 4 | Created in this study | | Details found in 'Preparation of rFel d 4' |
| Sequence-based reagent | Template plasmid DNAs for V2R cRNA probe | Designed in this study | | Sequence available in **Supplementary file 2** |
| Commercial assay or kit | ACTH ELISA kit | MD Bioproducts | # M046006 | |
| Commercial assay or kit | RNAscope Multiplex Fluorescent Reagent Kit v2 | ACD | #323100 | |
| Chemical compound, drug | 4-Hydroxytamoxifen | Sigma-Aldrich | #H6278 | |

## Animals

The experimental procedures were approved by the Institutional Animal Care and Use Committee at the University of California, Riverside (UCR) and were conducted in accordance with the National Institutes of Health Guide for the Care and Use of Laboratory Animals. C57BL/6N mice originally obtained from Charles River (Cat# 027) were bred at the UCR vivarium. The Trpc2 knock-out mouse line was generously provided by Dr. Richard Axel Lab at Columbia University. *Fos^{2A-iCreERT2}* (#030323)

and Ai9 (#007909) mice were originally obtained from The Jackson Laboratory and were bred at the UCR vivarium. $Fos^{2A-iCreERT2}$ mice were back-crossed with C57BL/6N strains of mice more than seven generations and bred with Ai9 mice to generate $Fos^{2A-iCreERT2}$; Ai9. All mice were housed in the vivarium under a reversed light cycle (lights on at 21:00; lights off at 9:00) for a minimum of 2 weeks prior to the behavior and histology experiments. Cats used in this study were obtained from the Nutrition and Pet Care Center at the University of California, Davis (a class A dealer) and were also housed in the UCR vivarium.

## Collection of saliva from cats

A person familiar with the cat gently held it for all procedures. A sterile cotton swab was inserted into the cat's mouth where the cat chewed and licked on the swab for at least 20 s or until the swab is saturated with saliva. The swabs containing saliva were then placed in tightly sealed microcentrifuge tubes and stored until used. Fresh saliva was collected within 15 min prior to its use in the mouse behavioral study, while old saliva was collected 4 hr prior to the study. The old saliva was kept at room temperature until it was used.

To measure the volume of saliva collected, a saliva-containing swab was placed onto a custom filter made from 0.5 mL and 1.5 mL microcentrifuge tubes and centrifuged at 2000 × $g$ for 1 min using a tabletop mini centrifuge. The protein concentrations of the eluted saliva were measured using the Bio-Rad Protein Assay (5000001, Bio-Rad).

## Western blot analysis

Proteins in saliva (10 μg protein) were separated by SDS-PAGE gels (456-1043, Bio-Rad, Hercules, CA), transferred onto PVDF membranes (162-0174, Bio-Rad), and incubated with rabbit anti-Fel d 4 antibody (1/1000: PA-FD4, Inbio). Bound antibody was detected by horseradish peroxidase-conjugated anti-rabbit IgG antibody (1/5000: 7074, Cell Signaling) and developed using Clarity Western ECL Substrate (1705060, Bio-Rad). Signals were detected and captured using ImageQuant LAS 4000 mini (GE Healthcare, Pittsburgh, PA), and band intensities were quantified with ImageJ software. Statistical significance was determined by unpaired $t$-test (two-tailed).

## Behavioral assay

All behavioral experiments were conducted during the animals' dark cycle. Mice were placed in their experimental room and allowed approximately 4 hr to acclimate to the ambient conditions. A cat saliva-containing swab was manually introduced into the animal's home cage to assess the defensive behavioral responses of resident mice. Recording of mouse behaviors commenced 2 min prior to introducing the sample and concluded 10 min thereafter. If tissues were required for histology, the animals were euthanized after 60 min, while for plasma collection, euthanasia took place after 15 min. Following the introduction of the sample, the behavioral room remained vacant of any individuals until the animal was euthanized. Mouse behavior was captured by night vision cameras equipped with infrared lights.

## Behavioral analysis

Recordings of each behavioral session were saved and provided in a blinded manner to a researcher for analysis. The analysis was performed using a behavioral analysis program called Behavioral Observation Research Interactive Software (*Friard and Gamba, 2016*). Freezing behavior was defined as immobility lasting more than 2 s and was scored accordingly. Stretch sniffing risk assessment behavior was scored when animals exhibited flat-back/stretch-attend posture. Direct investigation was scored when there was physical contact with the swab. Animals that exhibited inactivity in their home cage for more than 50% of the pre-drop analysis duration in each experiment were excluded from the data set.

## Histology and immunohistochemistry (IHC)

After being exposed to the cat specimen or control swab for 60 min, the animals were euthanized with an overdose of sodium pentobarbital. Intracardial perfusions were performed using 4% paraformaldehyde (PFA) in PBS. The brains were extracted and post-fixed in 4% PFA overnight at 4°C. Subsequently, the brains were cryoprotected by sequentially immersing them in 15 and 30% sucrose solutions in PBS. The VNOs were also extracted and underwent the same post-fixation procedure,

followed by decalcification with EDTA pH 8.0. Cryoprotection of the VNOs was achieved using a 30% sucrose solution in PBS. The cryoprotected tissues were then embedded in optimal cutting temperature (OCT) media and frozen over liquid nitrogen. Tissues embedded in OCT were sectioned at 14 μm (VNO) and 20 μm (brain) thicknesses using a cryostat set at –25°C. The sections were placed onto positively charged microscope slides. The slides were stored at –80°C until stained. A standard frozen IHC protocol was employed. For VNO sections, staining was performed using a rabbit anti-pS6 antibody (Cell Signaling Technology, #4858) at a dilution of 1:200, incubated overnight at 4°C, followed by a secondary antibody, Goat anti-rabbit IgG Alexa Fluor 647 (Life Technologies #a21245), for 60 min at room temperature. For AOB sections, staining was carried out using a mouse anti-Gαi2 antibody (Millipore #MAB3077MI) and a guinea pig anti-cFos antibody (Synaptic Systems #226 308) at dilutions of 1:500 and 1:1000, respectively. This was followed by incubation with a goat anti-mouse IgG2b Alexa Fluor 546 (Life Technologies #a21143) and a goat anti-guinea pig IgG Alexa Fluor 647. For brain sections, staining was performed using a guinea pig anti-cFos antibody at dilutions of 1:1000 incubated overnight at 4°C, followed by goat anti-guinea pig IgG Alexa Fluor 647. Finally, the slides were mounted using MS Shield Mounting Medium With 4,6-diamidino-2-phenylindole (DAPI) and DABCO (EMS #17989-20).

## Enzyme-linked immunosorbent assay (ELISA)

Plasma collected from mice exposed to cat saliva or control swabs was frozen at –80°C before initiating the ELISA protocol. The ACTH ELISA kit (MD Bioproducts) was utilized, following the instructions provided with the kit.

## ISH and dual staining

Detection of *GαO* RNA in fixed frozen VNO sections was conducted using the ACD RNAscope probe targeting GNAO1 (GαO) (ACD #444991) and the RNAscope Multiplex Fluorescent Reagent Kit v2 (ACD #323100). Probe binding was visualized using Akoya Biosciences' Opal 570 (FP1488001KT) Dye at a dilution of 1:750 in RNAscope TSA Buffer. Following RNAscope staining, the slides were incubated in anti-pS6 antibody at 1:200 dilution overnight at 4°C, then detected by goat anti-rabbit IgG Alexa Fluor 647 for 1 hr at room temperature.

Vmn2r (V2R) probes for the following genes were used: *Vmn2r13* for subfamily A1, *Vmn2r118* for A2, *Vmn2r117* for A3, *Vmn2r42* for A4, *Vmn2r69* for A5-1, *Vmn2r65* for A5-2, *Vmn2r76* for A5-3, *Vmn2r120* for A6, *Vmn2r98* for A8-1, *Vmn2r105* for A8-2, *Vmn2r58* for A8-3, *Vmn2r62* for A8-4, *Vmn2r81* for A9, *Vmn2r18* for A10, *Vmn2r24* for B, and *Vmn2r53* for D (*Supplementary file 2*). Each cRNA probe was synthesized using the PCR-based method (*Hua et al., 2018*) and plasmid DNAs containing probe sequences.

For the detection of *V2R* RNA in ISH, VNO tissues were sectioned and placed onto slides up to 1 week before ISH and stored at –80°C. Fluorescent ISH was performed following published protocols (*Ishii et al., 2004*). Briefly, the VNO slides were baked at 60°C for 5 min and fixed with 4% PFA. The slides were then treated with 10 μg/mL Proteinase K (Sigma #3115828001), followed by being acetylated in 0.1 M triethanolamine (Sigma #90279)-HCl with acetic anhydride (Sigma #320102). *V2R* cRNA probes were hybridized with tissue sections overnight at 65°C in a hybridization solution consisting of 50% formamide, 10 mM Tris-HCl pH 8.0, 200 μg/mL yeast tRNA, 10% dextran sulfate, 1× Denhardt's, 600 mM NaCl, 0.25% SDS, 1 mM EDTA pH 8.0. After hybridization, the slides were washed with 2× SSC in 50% formamide for 30 min, then 2× SSC, 0.2× SSC, and 0.1× SSC in $H_2O$ for 20 min each at 65°C. The slides were then blocked with blocking reagent (PerkinElmer) and 0.5% peroxidase block sequentially. Anti-DIG-POD (MilliporeSigma #11207733910) was used to detect DIG-labeled probes. TSA-Cy3 Plus kit (Akoya #NEL744001KT) was used to detect anti-DIG antibody at 1:200. The slides were incubated with anti-pS6 antibody at 1:200 dilution overnight at 4°C, then detected by goat anti-rabbit IgG Alexa Fluor 647 for 1 hr at room temperature.

Following staining, the slides were mounted with MS Shield Mounting Medium with DAPI and DABCO.

## Image quantification

Fluorescent images were captured at either ×10 or ×20 magnification using the Zeiss Axio Imager. M2 microscope. Whole-slide images of IHC-stained brains were scanned through fluorescent imaging

using Pannoramic SCAN (3D Histech) at Reveal Biosciences (San Diego, CA). Automated counting of fluorescent positive cells was performed using a proprietary code developed in our laboratory within the QuPath software.

### VMH cFos IR quantification and distribution analysis

All VMH regions were identified by clusters of nuclei visualized through DAPI staining and manually annotated in QuPath using standard coronal mouse brain atlas as reference. Annotations were processed with a proprietary script for immunoreactivity quantification. Distribution analyses were performed in QuPath using a rectangular annotation set parallel to the dorsomedial-ventrolateral axis of each VMH region. This rectangular annotation was then split into 10 equal bins, and immunoreactivity quantification in each bin was automated in QuPath, where the detections were finally manually recorded.

### Double exposure experiment

All behavioral experiments were conducted during the animals' dark cycle. Individual $Fos^{2A-iCreERT2}$; Ai9 mice in their home cages were placed in an experimental room and allowed to acclimate to the ambient conditions for approximately 4 hr. The 4-hydroxytamoxifen (4-OHT) solution was prepared as previously described (*DeNardo et al., 2019*). During the first exposure session, mice were injected intraperitoneally with 50 mg/kg of 4-OHT, 30 min prior to the introduction of the saliva sample. The first swab was removed from the cage 10 min after a mouse made the initial direct interaction with it. The mouse then remained in the cage for an additional 90 min before being returned to the vivarium. The second exposure session was conducted 1 week after the first session. For the second exposure, the mouse was kept with a swab for 60 min before being prepared for histological analysis. The behavioral room remained unoccupied during all exposure periods.

### Preparation of rFel d 4

cDNA of Fel d 4 was amplified from the pMAL-c2x-Fel d 4 plasmid (*Papes et al., 2010*), which was generously provided by Drs. Lisa Stowers and Fabio Papes. The Fel d 4 sequence was then subcloned into the expression vector pET-28a(+), and the His-tagged recombinant Fel d 4 (rFel d 4) was expressed in *Escherichia coli* BL21 (DE3). The rFel d 4 was purified using HisPur Ni-NTA Resin (Thermo Fisher Scientific) and dialyzed against PBS.

### Statistical analysis

Statistical analyses were performed using Prism software (GraphPad) or MATLAB (MathWorks). The sample size (N) for behavior experiment was based on $\alpha = 0.05$ and power = 0.8, using previously published variances for similar studies, our pilot experiments, and a 25% change as a relevant biological difference. Behavioral analysis results and cFos-IR distribution in the VMH, involving two variables, were compared using a traditional two-way ANOVA with Tukey's multiple comparisons. Behavioral and immunohistological results were statistically compared using one-way ANOVA with Tukey's multiple comparisons, assuming a parametric normal distribution. The correlation between immunoreactivity and behavior was analyzed using either Spearman's rank correlation coefficient or Pearson correlation coefficient: for data points that did not pass Shapiro–Wilk and Kolmogorov–Smirnov normality tests, Spearman's rank correlation coefficient was used, while for data points that pass Shapiro–Wilk and Kolmogorov–Smirnov normality tests, Pearson correlation coefficient was used. Statistical comparisons of activated neural populations in the double exposure experiment were conducted using permutation tests and chi-square tests.

## Acknowledgements

We appreciate the generosity of Dr. Michael Rust for providing a vivarium cat colony, Drs. C Ron Yu and Richard Axel for providing Trpc2 knock-out mice, and Drs. Lisa Stowers and Fabio Papes for providing Fel-d-4 plasmid DNA. We greatly appreciate Dr. Weifeng Gu for technical guidance on recombinant protein production. We thank Drs. C Ron Yu, Jing Wang, Martin M Riccomagno, Edward Zagha, Takuya Osakada, and Naoki Yamanaka for helpful discussion and comments on the manuscript. We are grateful for technical assistance from the Office of Campus Veterinarian at University of California, Riverside (UCR). This study was supported by funds from the National Institutes of Health

(R01DC019135) and startup funds from UCR to SHY, and a fellowship from Graduate Assistance in Areas of National Need to QAN.

## Additional information

### Funding

| Funder | Grant reference number | Author |
| --- | --- | --- |
| U.S. Department of Education | GAANN | Quynh Anh Thi Nguyen |
| National Institute on Deafness and Other Communication Disorders | R01DC019135 | Sachiko Haga-Yamanaka |

The funders had no role in study design, data collection and interpretation, or the decision to submit the work for publication.

### Author contributions

Quynh Anh Thi Nguyen, Formal analysis, Funding acquisition, Investigation, Methodology, Writing - original draft, Writing - review and editing; Andrea Rocha, Ricky Chhor, Yuna Yamashita, Christian Stadler, Crystal Pontrello, Investigation; Hongdian Yang, Formal analysis; Sachiko Haga-Yamanaka, Conceptualization, Formal analysis, Supervision, Funding acquisition, Investigation, Methodology, Writing - original draft, Writing - review and editing

### Author ORCIDs

Christian Stadler ⬤ https://orcid.org/0009-0000-7622-819X
Hongdian Yang ⬤ https://orcid.org/0000-0002-5203-9519
Sachiko Haga-Yamanaka ⬤ https://orcid.org/0000-0002-4101-9889

### Ethics

This study was performed in strict accordance with the recommendations in the Guide for the Care and Use of Laboratory Animals of the National Institutes of Health. All of the animals were handled according to approved institutional animal care and use committee (IACUC) protocols of the University of California, Riverside (UCR). The protocol was approved by the IACUC of the UCR (Institutional Lab Animal Assurance: D16-00278 (A3439-01), and AAALAC Accreditation: 000637).

Reviewer #1 (Public review): https://doi.org/10.7554/eLife.92982.4.sa1
Reviewer #2 (Public review): https://doi.org/10.7554/eLife.92982.4.sa2
Reviewer #3 (Public review): https://doi.org/10.7554/eLife.92982.4.sa3
Author response https://doi.org/10.7554/eLife.92982.4.sa4

## Additional files

### Supplementary files

• Supplementary file 1. Statistical summary of Tukey's multiple comparisons for *Figure 1K* and the two-way ANOVA with Tukey's multiple comparisons for *Figure 6B*.

• Supplementary file 2. Sequence of cRNA probes for V2R subfamilies.

• MDAR checklist

### Data availability

Figure 1—source data 1, Figure 2—source data 1, Figure 4—source data 1, Figure 5—source data 1, Figure 6—source data 1, and Figure 7—source data ! contain the numerical data used to generate the figures.

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
