## [Editor Report · eLife Assessment]

This **valuable** study addresses one way in which animals identify predator-associated cues and respond in a manner that reflects the imminence of the potential threat. The report shows that, in mice, fresh saliva from a natural predator (cat) elicits a greater defensive response compared to old cat saliva and implicates the vomeronasal organ and ventromedial hypothalamus as part of a circuit that underlies this process. The evidence supporting the main conclusions is **solid**. This study will be of interest to those interested in aversive behavior, its processes, and mechanisms.

---

## [Referee Report · Reviewer #1 (Public review)]

Summary:

Animals in natural environments need to identify predator-associated cues and respond with the appropriate behavioral response to survive. In rodents, some chemical cues produced by predators (e.g., cat saliva) are detected by chemosensory neurons in the vomeronasal organ (VNO). The VNO transmits predator-associated information to the accessory olfactory bulb, which in turn projects to the medial amygdala and the bed nucleus of the stria terminalis, two regions implicated in the initiation of antipredator defensive behaviors. A downstream area to these two regions is the ventromedial hypothalamus (VMH), which has been shown to control both active (i.e., flight) and passive (i.e, freezing) antipredator defensive responses via distinct efferent projections to the anterior hypothalamic nucleus or the periaqueductal gray, respectively. However, whether differences in predator-associated sensory information initially processed in the VNO and further conveyed to the VMH can trigger different types of behavioral responses remained unexplored. To address this question, here the authors investigated the behavioral responses of mice exposed to either fresh or old cat saliva, and further compared the underlying neural circuits that are activated by cat saliva with different freshness.

The scientific question of the study is valid, the experiments were well-performed, and the statistical analyses are appropriate. However, there are some concerns that may directly affect the main interpretation of the results.

In this revised version of the manuscript, the authors have made important modifications in the text, inserted new experiments and performed additional data analyses, as recommended. These modifications have significantly improved the quality of the manuscript and addressed all the major concerns detected during the prior submission.

---

## [Referee Report · Reviewer #2 (Public review)]

In this study, Nguyen et al. showed that cat saliva can robustly induce freezing behavior in mice. This effect is mediated through accessory olfactory system as it requires physical contact and is abolished in Trp2 KO mice. The authors further showed that V2R-A4 cluster is responsive to cat saliva. Lastly, they demonstrated c-Fos induction in AOB and VMHdm/c by the cat saliva. The c-Fos level in the VMHdm/c is correlated with freezing response.

Strength:

The study opens an interesting direction. It reveals the potential neural circuit for detecting cat saliva and driving defense behavior in mice. The behavior results and the critical role of accessory olfactory system in detecting cat saliva are clear and convincing.

Weakness:

The findings are relatively preliminary. The identities of the receptor and the ligand in the cat saliva that induces the behavior remain unclear. The identity of VMH cells that are activated by the cat saliva remains unclear. There is a lack of targeted functional manipulation to demonstrate the role of V2R-A4 or VMH cells in the behavioral response to the cat saliva.

Here are some specific comments:

(1) This result suggests that V2R-A4 may be the dominant VR for mice to detect cat saliva. Future studies should determine the identity of the receptor and the ligand in the cat saliva. Additionally, the functional importance of V2R-A4 remains unclear. It is important to knockout the receptor and test changes in cat saliva-induced freezing.

(2) AOB does not project to VMH directly. Other known important nodes for the predator defense circuit includes MeApv, BNST, PMd, AHN and PAG. It will be helpful to provide c-Fos data in those regions (especially MEA and BNST as they are between AOB and VMH) to provide a complete picture regarding how the brain process cat saliva to induce the behavior change.

(3) It is interesting that activation level difference in the VNO by old and fresh cat saliva does not transfer to AOB. It could be informative to examine correlation between VNO and AOB p6/c-Fos cell number and AOB and VMH c-Fos cell number across animals to understand whether the activation level across those regions are related. If they are not correlated, it could be helpful to add a discussion regarding potential reasons, e.g. neuromodulatory inputs to the AOB.

(4) Please indicate n in all figure plots and specify what individual dots means. In Figure 4h, there are 7 dots in old saliva group, presumably indicating 7 animals. In Figure 6b, there appear to be more than 7 dots for old cat saliva group. Are there more than 7 animals? If so, why are they not included in Figure 4h? If not, what does each dot mean? Note that each dot should represent independent sample. One animal should not contribute more than one dot.

(5) The identification of a cluster of VMHdm cells uniquely activated by fresh cat saliva urine is interesting. It will be important to identify the molecular handle of the cells to facilitate further investigation. This could be achieved using either activity dependent RNAseq or double in situ of saliva-induced c-Fos and candidate genes (candidate gene may be identified based on the known gene expression pattern).

---

## [Referee Report · Reviewer #3 (Public review)]

Summary:

Nguyen et al show data indicating that the vomeronasal organ (VNO) and ventromedial hypothalamus (VMH) are part of a circuit that elicits defensive responses induced by predator odors. They also suggest that using fresh or old predator saliva may be a method to change the perceived imminence of predation. The authors also identify a family of VNO receptors that are activated by cat saliva. Next, the authors show how different components of this defensive circuit are activated by saliva, as measured by fos expression. The work also shows that different VMH populations are activated by fresh and old saliva, demonstrating that these stimuli create qualitatively different neural activity profiles. However, the exact components that differ between fresh and old saliva remain unknown and may be identified in future studies.

Strengths:

(1) Predator saliva is a stimulus of high ethological relevance

(2) The authors performed a careful quantification of fos induction across the anterior-posterior axis

(3) Authors show that different VMH populations are activated by fresh and old saliva

Weaknesses:

(1) There is a lack of standard circuit dissection methods, such as characterizing the behavioral effects of increasing and decreasing neural activity of relevant cell bodies and axonal projections

(2) Some of the findings are disconnected from the story. For example, the authors show V2R-A4-expressing cells are activated by predator odors, but the causal role of these cells in generating defensive actions is not shown

---

## [Author Response]

The following is the authors’ response to the previous reviews

We greatly appreciate all the reviewers’ constructive comments on our previously revised manuscript. In the current revision, we added several experimental data for answering the reviewers’ comments. Below we describe our point-by-point responses to their comments:

**Reviewer #1 (Public Review):**
Unaddressed and additional concerns (re-submission)In this revised version of the manuscript, the authors have made important modifications in the text, inserted new references, and incorporated additional quantifications of cFos immunolabeling in three brain regions, as recommended by the reviewers. While these modifications have significantly improved the quality of the manuscript, other critical concerns raised during the initial submission of themanuscript (Major concerns 1, 2, and 4; some of them also raised by the other reviewers) were not properly addressed by the authors. On several occasions, the authors recognize the importance of clarifying the points for the correct interpretation of the results but opt for leaving the open questions to be addressed during future studies. Therefore, the authors might consider adding a new section at the end of the manuscript to include all the caveats and future directions.

In the current revision, in order to answer the reviewer #1’s original concerns 1, 2, and 4, we added several experimental data.

Original major concerns (1) and (2): Regarding whether mice are detecting qualitative or quantitative differences between fresh and old cat saliva.

To address these concerns, as shown in new Figure 1I and J, we measured volumes of saliva contained in in individual swabs and total protein concentrations at the time of behavior tests: Fresh (15 minutes after collection) and Old (4 hours after collection). The saliva volumes at the time of behavioral testing were indistinguishable between fresh and old samples (Figure 1I). In addition, the concentrations of total proteins in both fresh and old saliva were also indiscernible (Figure 1J). Furthermore, we also examined the difference of the amount of Fel d 4 protein, one of the most abundant proteins in cat saliva, between fresh and old saliva by conducting western blotting analyses. As shown in new Supplemental Figure 2, the amount of Fel d 4 was nearly equivalent between fresh and old saliva. Indeed, our analyses using recombinant Fel d 4 protein showed that Fel d 4 does not induce freezing behavior (Supplemental Figure 5). Based on these findings, we believe that the difference between fresh and old cat saliva lies in specific components rather than the total or major saliva content. One possible explanation for this difference is the time-dependent reduction of specific freezing-inducing components in old saliva.

To investigate such a possibility, we also examined mouse behavior directed toward swabs containing diluted fresh cat saliva. Indeed, exposure to diluted fresh saliva resulted in a shorter duration of freezing behavior. Fresh saliva diluted to 70% induced freezing behavior for a duration equivalent to that of undiluted fresh saliva, while freezing behavior in response to 50% and 30% fresh saliva was significantly reduced to the same duration as that observed with old saliva (Figure 1K). The duration of direct interaction with swabs containing 70% and 50–30% fresh saliva also exhibited a similar trend to that observed with fresh and old saliva swabs, respectively (Figure 1L).

These new results provide compelling evidence that the differential freezing response of mice to fresh versus old cat saliva is not attributed to quantitative differences, such as total volume, total protein concentration, or the amount of major proteins like Fel d 4. However, when fresh saliva was diluted, we observed a corresponding reduction in freezing behavior, suggesting that specific components within the saliva—those responsible for inducing freezing—may decrease over time.

Our findings indicate that while the overall content of saliva remains consistent over time, specific freezing-inducing components seem to degrade or reduce at a different rate than other components, which alters the composition of saliva over time. The speed of reduction of these freezing-inducing components appears to be different from more stable proteins such as Fel d 4. As a result, the composition of saliva changes over time, leading to a qualitative difference between fresh and old saliva that mice can detect. This ability to discern such subtle chemical changes likely reflects an adaptive sensory mechanism, allowing mice to respond to predator cues to induce optimal defensive behavior in a certain context. Identifying the specific freezing-inducing components through traditional purification processes, such as high-performance liquid chromatography followed by behavioral examination (Haga-Yamanaka et al., 2014; Kimoto et al., 2005), is crucial for a deeper understanding of the mechanisms underlying the observed behavior. Our research team is actively working to isolate these molecules, and we hope to report our findings in future studies.

(4) The interpretation that fresh and old saliva activates different subpopulations of neurons in the VMH based on the observation that cFos positively correlates with freezing responses only with the fresh saliva lacks empirical evidence. To address this question, the authors should use two neuronal activity markers to track the response of the same population of VHM cells within the same animals during exposure to fresh vs. old saliva.

To address this issue, as shown in the new Figure 7, we performed a double exposure experiment using Fos2A-iCreERT2; Ai9 (TRAP2) mice (Allen et al., 2017; DeNardo et al., 2019). In this experiment, mice were exposed to the first stimulus under the treatment of 4-hydroxytamoxifen (4-OHT). One week after the initial exposure, the same mice were subjected to a second stimulus exposure for one hour. Through this paradigm, neurons activated by the first stimulus were visualized by tdTomato, while ones activated by the second stimulus were detected as cFos-IR (Figure 7A). Quantification of tdTomato and cFos-IR double-positive cells among tdTomato-labeled cells revealed that 43% (mean per animal: 61 / 143) of cells activated by fresh saliva during the first exposure were also activated by fresh saliva during the second exposure, whereas only 16% (17 / 106) of cells activated by old saliva during the first exposure were activated by fresh saliva during the second exposure (p = 7.5e-6, Chi-squared test). The difference in the fraction of overlapping cells between fresh and old saliva exposures was found significant when we compared the two groups of animals (Figure 7D, p = 0.0035, permutation test). Additionally, quantification of tdTomato and cFos-IR double-positive cells among cFos-IR cells indicated that over 27% (61 / 226) of cells activated by fresh saliva during the second exposure were previously activated by fresh saliva, whereas only 15% (17 / 112) of cells activated by fresh saliva during the second exposure were previously activated by old saliva (p = 0.015, Chi-squared test). The difference in the fraction of overlapping cells between fresh and old saliva exposures was also significant in this analysis (Figure 7E,p = 0.0060, permutation test). Together, these results demonstrate that fresh and old cat saliva activate largely different populations of neurons within the VMH. These new results were described on page 11 line 18 – page 12 line 8.

In addition to these unaddressed concerns, some new issues have emerged in the new version of the manuscript. For example, the following paragraph introduced in the discussion section is not supported by the experimental findings."We assume that such differential activations of the mitral cells between fresh and old saliva result in the differential activation of targeting neural substrates, possibly MeApv, which results in differential activation of VMH neurons (Figure 7)."Although the authors did not observe statistical differences in cFos expression in the pvMeA among groups, they claim that the differences in cFos expression in the VMH between fresh vs. old saliva are mediated by differential activation of upstream neurons in the MeApv. The lack of statistical differences may be caused by the reduced number of subjects in each group, as recognized in the text by theauthors.

We appreciate the reviewer's thoughtful comment. We agree that the paragraph in the comment, which presented a working hypothesis regarding differential activations of mitral cells and the MeApv between fresh and old saliva exposures, was speculative and not fully supported by our experimental findings. To address this, we have removed the assumptions related to the differential responses of mitral cells and the MeApv from the discussion and have updated the figure accordingly (now presented as new Figure 8).

Moreover, the authors propose that in addition to fel d 4, multiple molecules present in the cat saliva can be inducing distinct defensive responses in the animals, but they do not provide any reference to support their claim.

We thank the reviewer for highlighting this point. Our claim regarding the presence of other molecules in cat saliva inducing freezing defensive responses is based on our observation, as shown in the new Supplemental Figure 5, that recombinant Fel d 4 protein alone does not induce freezing behavior. This suggests the existence of other unidentified components in cat saliva that may contribute to freezing behavior. As we agree that identifying these specific freezing-inducing components is important for a more comprehensive understanding of the underlying mechanisms, our research team is actively working to isolate these molecules, and we hope to report our findings in future studies.

**Reviewer #2 (Public Review):**
The findings are relatively preliminary. The identities of the receptor and the ligand in the cat saliva that induces the behavior remain unclear. The identity of VMH cells that are activated by the cat saliva remains unclear. There is a lack of targeted functional manipulation to demonstrate the role of V2R-A4 or VMH cells in the behavioral response to the cat saliva.

We thank the reviewer’s important insight on the need for further investigation into the molecular and neural mechanisms underlying the behavioral response to cat saliva. We recognize the importance of conducting studies involving V2R-A4 receptor knockouts and targeted functional manipulations within the VMH using neural circuit perturbation approaches.

However, the V2R-A4 subfamily consists of 25 Vmn2r genes, most of which are closely grouped together, forming a V2R-A4 gene cluster within a 2.5-megabase chromosomal region. As we described in our recent review article (Rocha et al., 2024), the Vmn2r genes within the V2R-A4 subfamily display a high degree of homology, with nucleotide and amino acid identities among the several Vmn2rs surpassing 97-99%, suggesting possible redundancy among these receptor genes. This is in stark contrast to the diversity typically observed within other V2R subfamilies. Consequently, knockout strategies targeting a single receptor gene, which have been successful for other vomeronasal receptors, may not be effective for V2R-A4 receptor genes. The most appropriate strategy for examining the necessity of V2R-A4 receptors would be knocking out the entire V2R-A4 gene cluster, spanning a 2.5-megabase chromosomal region. Due to the technical challenges involved, addressing this issue is not feasible in the foreseeable future. Moreover, in our current study, we aimed to establish the foundational relationship between predator cues in cat saliva and defensive behaviors. We view our findings as an important first step that sets the stage for these more targeted and mechanistic studies involving the neural circuit perturbation experiments, such as optogenetics and Designer Receptors Exclusively Activated by Designer Drugs (DREADDs), in the next step.

**Reviewer #3 (Public Review):**
Weaknesses:(1) It is unclear if fresh and old saliva indeed alter the perceived imminence of predation, as claimed by the authors. Prior work indicates that lower imminence induces anxiety-related actions, such as re- organization of meal patterns and avoidance of open spaces, while slightly higher imminence produces freezing. Here, the authors show that fresh and old predator saliva only provoke different amounts of freezing, rather than changing the topography of defensive behaviors, as explained above. Another prediction of predatory imminence theory would be that lower imminence induced by old saliva should produce stronger cortical activation, while fresh saliva would activate amygdala, if these stimuli indeed correspond to significantly different levels of predation imminence.

We appreciate the reviewer’s insightful comments regarding the perceived imminence of predation and the behavioral responses to fresh and old saliva. Our study specifically focused on comparing the defensive behaviors of mice in response to 15-minute-old and 4-hour-old cat saliva, particularly within the context of freezing behavior in their home cages. We chose these specific time points to capture the potential variation in behavioral intensity rather than the full spectrum of defensive behaviors. While a more comprehensive analysis—including varying time points, different types of defensive behaviors, and broader neural activation patterns (e.g., cortical versus amygdala activation)—might provide further insights into predation imminence theory, these aspects were beyond the scope of our current study. Future research could certainly address these points by examining behavioral and neural responses across additional saliva aging intervals and in varied behavioral contexts. Such studies would complement and extend the findings presented here, further elucidating the relationship between predator cue characteristics and defensive behaviors.

(2) It is known that predator odors activate and require AOB, VNO and VMH, thus replications of these findings are not novel, decreasing the impact of this work.

As the reviewer mentioned, the activation of the AOB, VNO, and VMH by predator odors has been established in prior studies. However, our study provides new insights by demonstrating that defensive freezing behavior in response to predator odors is mediated through the vomeronasal organ (VNO) sensory circuit, which has not been previously shown. The novelty of our work lies in two key findings: (1) the introduction of a new behavioral paradigm that assesses freezing responses to predator cues based on the freshness of chemosensory signals in cat saliva, and (2) the demonstration that the vomeronasal sensory circuit mediates defensive freezing behavior in response to cat saliva.

Additionally, our results show that cat saliva of different freshness levels differentially activates VNO sensory neurons that express the same subfamily of sensory receptors. This differential activation subsequently modulates the downstream neural circuits, leading to varied freezing behavioral outcomes. We believe these findings provide a novel conceptual advance over previous studies by elucidating a more detailed mechanism of how predator-derived cues influence defensive behaviors through the accessory olfactory system.

(3) There is a lack of standard circuit dissection methods, such as characterizing the behavioral effects of increasing and decreasing neural activity of relevant cell bodies and axonal projections, significantly decreasing the mechanistic insights generated by this work

We thank the reviewer for this valuable comment. Investigating the behavioral effects of manipulating specific cell types and axonal projections, as well as characterizing circuit connectivity, is essential for a more comprehensive understanding of the underlying neural circuits. These approaches, such as modulating neural activity in defined cell populations and dissecting circuit pathways, using optogenetics, DREADD, etc., would provide deeper mechanistic insights. In our current study, however, we aimed to establish the foundational relationship between predator cues in cat saliva and defensive behaviors. We view our findings as an important first step that sets the stage for these more targeted and mechanistic studies in the future.

(4) The correlation shown in Figure 5c may be spurious. It appears that the correlation is primarily driven by a single point (the green square point near the bottom left corner). All correlations should be calculated using Spearman correlation, which is non-parametric and less likely to show a large correlation due to a small number of outliers. Regardless of the correlation method used, there are too few points in Figure 5c to establish a reliable correlation. Please add more points to 5c.

We appreciate the reviewer’s suggestion regarding the correlation analysis in Figure 5E. We assessed the normality of our data using both the Shapiro-Wilk and Kolmogorov-Smirnov tests, which confirmed that the dataset is parametric, justifying the use of a parametric correlation method in this context. However, we acknowledge the concern about the limited number of data points and the influence of potential outliers on the observed correlation. Increasing the sample size might provide a more robust assessment of correlation patterns and reduce the potential impact of any single data point. While this would be an important direction for future research, such as with larger sample sizes, it is beyond the scope of the current study.

(5) Please cite recent relevant papers showing VMH activity induced by predators, such as https://pubmed.ncbi.nlm.nih.gov/33115925/ and https://pubmed.ncbi.nlm.nih.gov/36788059/

We thank the reviewer’s suggestion to cite these important papers. https://pubmed.ncbi.nlm.nih.gov/33115925/ (Esteban Masferrer et al., 2020) and https://pubmed.ncbi.nlm.nih.gov/36788059/ (Tobias et al., 2023) are now cited at page 16 line 10 in Discussion under “Differential activation of VMH neurons potentially underlying distinct intensities of freezing behavior.”

(6) Add complete statistical information in the figure legends of all figures, which should include n, name of test used and exact p values.

We included statistical analysis results in figure legends; for Figure 6B, we provided statistical analysis results in Supplemental Table 1.

(7) Some of the findings are disconnected from the story. For example, the authors show V2R-A4- expressing cells are activated by predator odors. Are these cells more likely to be connected to the rest of the predatory defense circuit than other VNO cells?

Yes, our hypothesis posits that V2R-A4-expressing VNO sensory neurons serve as receptor neurons for predator cues present in cat saliva. Additionally, we assume that these specific sensory neurons have stronger anatomical connections with the defensive circuit compared to VNO sensory neurons expressing other receptor subfamilies. In our modified Discussion section, we discussed this point under “V2R-A4 subfamily as the receptor for predator cues in cat saliva.”

(8) Please paste all figure legends directly below their corresponding figure to make the manuscript easier to read

We have added figure legends directly below their corresponding figures.

(9) Were there other behavioral differences induced by fresh compared to old saliva? Do they provoke differences in stretch-attend risk evaluation postures, number of approaches, average distance to odor stimulus, velocity of movements towards and away the odor stimulus, etc?

We appreciate the reviewer's valuable comments. We have now incorporated an analysis of stretch-sniff risk assessment behavior, presented in new Figure 1F (graph) and Supplemental Figure 1B (raster plot). Mice exhibited stretch-sniff risk assessment behavior, which remained consistent across control, fresh saliva, and old saliva swabs. Additionally, we have also included a raster plot for direct investigation, previously noted as ‘interaction’ in the original manuscript (Supplemental Figure 1C). Mice exposed to a swab containing either fresh or old saliva significantly avoided directly investigating the swab. In contrast, mice exposed to a clean control swab spent a significant amount of time directly investigating the swab, engaging in behaviors such as sniffing and chewing (Figure 1G). A comparison of temporal behavioral patterns revealed a slightly higher frequency of direct investigation behavior toward old saliva compared to fresh saliva at the beginning of the exposure period (Supplemental Figure 1C).

**Reviewer #3 (Recommendations For The Authors):**
The authors have partially addressed several important points raised in the prior review, increasing the strength of the manuscript. However, 2 key questions already raised previously, were not addressed:(1) Is old saliva qualitatively different from new saliva, or is it the same as a smaller amount of new saliva? As Reviewer 1 wrote: "An important point that the authors should clarify in this study is whether mice are detecting qualitative or quantitative differences between fresh and old cat saliva."Since one of the author's main points is that fresh and old saliva elicit different perceived threat imminences, it is crucial to show that these two stimuli are somehow qualitatively different.One way to investigate this could be to show that animals perform different behaviors when exposed to smaller among of new saliva vs. old saliva, or that the cfos activation patterns are different in these two conditions.

The answers to these concerns are provided in the Public review Comment from Reviewer #1.

(2) The other key question is if different VMH populations are activated by new vs. old saliva.

The answer to this concern is provided in the Public Review comment from Reviewer #1.

Lastly, although the new analysis and text changes improved the manuscript, many issues raised were addressed with some variation of 'future studies will be done', or 'we concur with the Reviewer'. However, the extra experiments required to answer these questions were not done. For this reason, even though the authors have numerous exciting pieces of data, overall the work is still incomplete. I highlight below some examples in which the authors agree with the Reviewer, but do not answer the question with the new work that would be required, or propose to do the work in future studies.

In this revised manuscript, we have conducted several additional experiments to address key concerns raised by the reviewers that are directly relevant to our claims. Specifically, we have examined: (1) whether qualitative or quantitative differences between fresh and old cat saliva are detected by mice to modulate behavior (NEW Figure 1I, J, K, and L, and NEW Supplemental Figure 2); (2) the involvement of Fel d 4 in freezing behavior (NEW Supplemental Figure 5); and (3) whether different VMH populations are activated by fresh versus old saliva (NEW Figure 7). However, some concerns raised by the reviewers fall outside the scope of the current manuscript. These include: (1) identifying the specific components that induce freezing, (2) examining the necessity of V2R-A4 receptors, (3) conducting neural circuit perturbations, and (4) performing a comprehensive analysis—including varying time points, different types of defensive behaviors, and broader neural activation patterns (e.g., cortical versus amygdala activation)—of the mouse’s defensive response to different levels of predator threat imminence. As these aspects are beyond the focus of our current manuscript, we have noted in the Public Review comments.

References:

Allen WE, DeNardo LA, Chen MZ, Liu CD, Loh KM, Fenno LE, Ramakrishnan C, Deisseroth K, Luo L. 2017. Thirst-associated preoptic neurons encode an aversive motivational drive. *Science* 357:1149– 1155.

DeNardo LA, Liu CD, Allen WE, Adams EL, Friedmann D, Fu L, Guenthner CJ, Tessier-Lavigne M, Luo L. 2019. Temporal evolution of cortical ensembles promoting remote memory retrieval. *Nat Neurosci* 22:460–469.

Haga-Yamanaka S, Ma L, He J, Qiu Q, Lavis LD, Looger LL, Yu CR. 2014. Integrated action of pheromone signals in promoting courtship behavior in male mice. *Elife* 3:e03025.

Kimoto H, Haga S, Sato K, Touhara K. 2005. Sex-specific peptides from exocrine glands stimulate mouse vomeronasal sensory neurons. *Nature* 437:898–901.

Rocha A, Nguyen QAT, Haga-Yamanaka S. 2024. Type 2 vomeronasal receptor-A4 subfamily: Potential predator sensors in mice. *Genesis* 62:e23597.